# TENSOR TIME-SERIES FORECASTING AND ANOMALY DETECTION WITH AUGMENTED CAUSALITY

## ABSTRACT

In time series, variables often exhibit high-dimensional characteristics. The correlation between variables tends to be intricate, encompassing aspects such as non-linearity and time dependency. Understanding the interaction of variables and comprehending the distribution of their values can significantly enhance the effectiveness of time series data analysis tasks, such as forecasting and anomaly detection. Hence, in this paper, we start from the tensor time series, which can encode higher dimensional information than classic multivariate time series, and aim to discover and leverage their fine-grained time-dependent causal relations to contribute to a more accurate analysis. To this end, we first form an augmented Granger Causality model, named TBN-Granger Causality, which adds time-respecting Bayesian Networks to the time-lagged Neural Granger Causality through a bi-level optimization, such that the overlooking of instantaneous effects in typical causal time series analysis can be addressed. Then, we propose an end-to-end deep generative model, named **TacSas**, which takes the historical tensor time series, outputs the future tensor time series, and detects possible anomalies by leveraging the TBN-Granger Causality in the history. Moreover, we show Tac-Sas not only can capture the ground-truth causality but also can be applied when the ground-truth causal structures are hardly available, to help forecasting and anomaly detection. For evaluations, besides synthetic benchmark data, we have four datasets from the climate domain benchmark database ERA5 as the real-world tensor time series for forecasting. Moreover, we extend ERA5 with the extreme weather database NOAA for testing anomaly detection accuracy. We show the effectiveness of TacSas in different time series analysis tasks by comparing causal baselines, forecasting baselines, and anomaly detection baselines.

## 1 INTRODUCTION

Time series analysis is indispensable in various application domains. Time-series forecasting, for example, can facilitate traffic planning (Li et al., 2018; Zhao et al., 2020b). Time-series anomaly patterns can optimize high-tech equipment deployment (Hundman et al., 2018; Su et al., 2019). In the real world, time series variables usually contain high-dimensional features. Taking the climate time series data as an example, multiple variables such as temperature, wind, atmospheric water content, and solar radiation co-appear on the time axis. Although we can access their tabular representations, their interactions are typically complex (e.g., non-linear, time-dependent), making it difficult to understand and capture the time series evolution trend and latent distribution of values. As a result, this complexity may lead to sub-optimal performance in time series analysis, such as time series forecasting and anomaly detection. Motivated by the above, **structured learning in time series** has recently gained much attention, such as (Li et al., 2018; Wu et al., 2020; Zhao et al., 2020a; Cao et al., 2020; Shang et al., 2021; Deng and Hooi, 2021; Marcinkevics and Vogt, 2021; Geffner et al., 2022; Tank et al., 2022; Spadon et al., 2022; Gong et al., 2023). Among others, causal graphs as a directed acrylic graph structure provide more explicit and interpretable correlations between variables, thus enabling a better understanding of the underlying physical mechanisms and dynamic systems for time series (Guo et al., 2021).

As a widely applied causal structure in time series understanding and explanation, Granger Causality (Granger, 1969; Arnold et al., 2007) discovers causal relations among variables in an autoregressive (or time-lagged) manner. The discovered Granger Causal structures can help many time series

analysis tasks, like building parsimonious prediction models such as Earth System (Runge et al., 2019). Moreover, real-world time series data can have many variables, and their causal relations can be more complex, i.e., non-linear and instantaneous, which require complex causality discovery beyond the classic Granger model. On the one hand, some nascent non-linear (or neural) Granger models have been proposed (Nauta et al., 2019; Xu et al., 2019; Tank et al., 2022; Khanna and Tan, 2020; Huang et al., 2020; Pamfil et al., 2020; Marcinkevics and Vogt, 2021; Geffner et al., 2022). On the other hand, how to effectively integrate instantaneous causal effects with neural Granger models has the great research potential (Moneta et al., 2013; Wild et al., 2010; Dahlhaus and Eichler, 2003; Malinsky and Spirtes, 2018; Assaad et al., 2022) but remains largely under-explored (Pamfil et al., 2020; Gong et al., 2023).

Motivated by the above analysis, in this paper, we start from the tensor time series data as shown in Figure 1(a), in which the 3D structure contains higher dimensions than typical 2D multivariate time series data. For example, tensor time series can represent multivariate climate time series data (e.g., temperature, wind, and atmospheric water content) with corresponding spatial information (e.g., longitude, latitude, and geocode). After that, we aim to build a comprehensive causality model for this tensor time series, which could not only capture non-linear and time-lagged causality (like the Granger model (Granger, 1969; Tank et al., 2022)) but also offset the ignored instantaneous causal

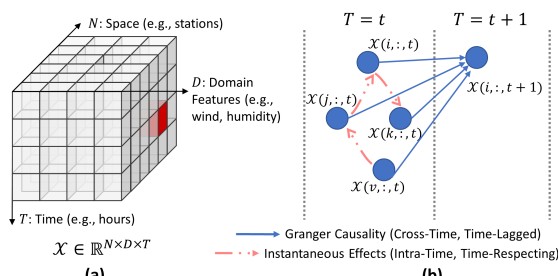

Figure 1: (a) Tensor Time-Series Data: The Red Cell Means the Possible Anomaly. (b) Visualization of (Neural) Granger Causality's Time-Lagged Property without Instantaneous Effects.

effects at each timestamp, as shown in Figure 1(b). Our **ultimate goal** is to leverage the discovered comprehensive causality to understand the trend and latent distribution of the historical tensor time series and finally contribute to the analysis tasks like tensor time series forecasting and anomaly detection.

To this end, we first propose a comprehensive causal model named Time-Respecting Bayesian Network Augmented Neural Granger Causality, i.e., TBN-Granger Causality. Theoretically, discovering TBN-Granger Causality relies on a bi-level optimization. The inner optimization discovers a sequence of Bayesian Networks at each timestamp $t$ respectively for representing the instantaneous causal effects among variables (i.e., which causality is responsible for the instantaneous feature generation). Then, the outer optimization realizes integrating time-respecting Bayesian Networks with time-lagged neural Granger causality in an autoregressive manner. Empirically, to embed TBN-Granger Causality into guiding the tensor time series analysis tasks like forecasting and anomaly detection, we propose an end-to-end deep generative model, called **T**ime-**A**ugmented **C**ausal Time **S**eries **A**nalysi**S** Model, i.e., TacSas.

Different from previous causal time series analysis works (Nauta et al., 2019; Xu et al., 2019; Khanna and Tan, 2020; Pamfil et al., 2020; Huang et al., 2020; Marcinkevics and Vogt, 2021; Tank et al., 2022; Geffner et al., 2022; Gong et al., 2023), TacSas takes a step further from merely verifying whether the ground-truth causal structures in a synthetic setting are discovered. Moreover, it also investigates how to capture good causal structures when the ground-truth structures are hardly available to guide time series analysis tasks further. Thus, TacSas adopts the generative learning manner, which does not need any labeled causal structures or time series. Furthermore, TacSas is end-to-end, meaning that it can not only discover TBN-Granger Calsuality from the observed time series, TacSas can but also seamlessly use it to forecast future time series and detect possible anomalies.

To evaluate TacSas, we first use the synthetic benchmark, Lorenz-96 (Lorenz, 1996), to verify that TacSas can indeed capture ground-truth causal structures with high accuracy. Then, we extend to the real-world setting and test if TacSas can utilize the captured causality to finish tensor time series forecasting and identify anomalies. We have four tensor time series datasets from the hourly climate benchmark database ERA5 (Hersbach et al., 2018) and align them with the weather anomaly database NOAA [1]. The results show that TacSas outperforms forecasting and detection baselines.

---

[1] https://www.ncdc.noaa.gov/stormevents/ftp.jsp

## 2 PRELIMINARY

**Tensor Time Series**. As shown in Figure 1(a), we have tensor time series data stored in $\mathcal{X} \in \mathbb{R}^{N \times D \times T}$. Note that a slice of $\mathcal{X}$, i.e., $\mathcal{X}(i, :, :) \in \mathbb{R}^{D \times T}$, $i \in \{1 \dots, N\}$, is typically denoted as the common multivariate time series data (Su et al., 2019; Zhao et al., 2020a). In this way, tensor time series can be understood as multiple multivariate time series data. Such tensor time series data can usually be found in the real world. For example, in each element $\mathcal{X}(i, d, t)$ of the nationwide weather data $\mathcal{X}$, $i \in \{1 \dots, N\}$ can be the spatial locations (e.g., counties), $d \in \{1 \dots, D\}$ can be the weather features (e.g., temperature and humidity), and $t \in \{1 \dots, T\}$ can be the time dimension (e.g., hours). Throughout the paper, we use the calligraphic letter to denote a 3D tensor (e.g., $\mathcal{X}$) and the bold capital letter to denote a 2D matrix (e.g., $\boldsymbol{X}$).

**Problem Definition**. In this paper, we aim to discover and utilize comprehensive causal structures for tensor time-series analysis tasks, including forecasting and anomaly detection. To be more specific, given the tabular data $\mathcal{X} \in \mathbb{R}^{N \times D \times T}$ as shown in Figure 1, we aim to forecast the future data $\mathcal{X}' \in \mathbb{R}^{N \times D \times \tau}$, where $\tau$ is a forecasting window. Additionally, with the forecasted $\mathcal{X}'$, we also aim to detect if $\mathcal{X}'$ contains abnormal values.

## 3 TACSAS: DISCOVERING TBN-GRANGER CAUSALITY FOR TENSOR TIME SERIES FORECASTS AND ANOMALY DETECTIONS

In this section, we introduce how TacSas discovers TBN-Granger Causality in the historical tensor time series and utilizes it to guide tensor time series forecasting and anomaly detection. The overall framework of TacSas is shown in Figure 2.

The upper component of Figure 2 represents the data preprocessing part (i.e., converting raw input $\mathcal{X}$ to latent representation $\mathcal{H}$) of TacSas through a pre-trained autoencoder. The goal of this component is leveraging comprehensive causality (e.g., TBN-Granger Causality) to achieve seamless forecasting and anomaly detection. The theoretical reasoning and necessity are introduced in Sec.3.3, and the empirical validation is demonstrated in Appendix B.2.

The lower component of Figure 2 shows how TacSas discovers TBN-Granger Causality in the historical tensor time series (in the form of $\mathcal{H}$ other than $\mathcal{X}$) and generates future tensor time series. In brief, the optimization of TacSas is bi-level. First, the inner optimization captures instantaneous effects among variables at each timestamp, respectively, which describes the inner-time feature generation. These causal structures are then stored in the form of a sequence of Bayesian Networks. The details are introduced in Sec.3.1. Second, the outer optimization discovers the Neural Granger Causality among variables in a time window with the support of a sequence of Bayesian Networks (i.e., TBN-Granger Causality). After introducing details in Sec.3.2, we derive the formal equation of TBN-Granger Causality, Eq. 3.7.

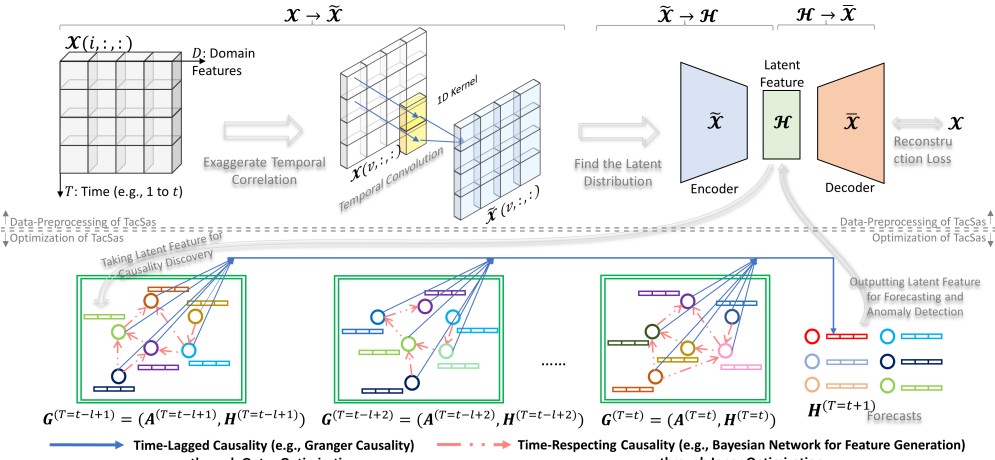

Figure 2: Proposed TacSas for Discovering and Utilizing TBN-Granger Causality in Tensor Time-Series Forecasting and Anomaly Detection.

## 3.1 INNER OPTIMIZATION OF TACSAS FOR IDENTIFYING INSTANTANEOUS CAUSAL RELATIONS IN TENSOR TIME SERIES

Generally speaking, the inner optimization produces a sequence of Bayesian Networks for each observed timestamp. At time $t$, the instantaneous causality is discovered based on input features $\mathcal{H}(:,:,t) = \boldsymbol{H}^{(t)} \in \mathbb{R}^{N \times H}$, and is represented by a directed acyclic graph $\mathcal{G}^{(t)} = (\boldsymbol{A}^{(t)} \in \mathbb{R}^{N \times N}, \boldsymbol{H}^{(t)} \in \mathbb{R}^{N \times H})$. To be specific, $\boldsymbol{A}^{(t)}$ is a weighted adjacency matrix of the Bayesian Network at time $t$, and each cell represents the coefficient of causal effects between variables $u$ and $v \in \{1, \ldots, N\}$. The features (e.g., $\mathcal{H}(v,:,t)$) are transformed from the input raw features (e.g., $\mathcal{X}(v,:,t)$). The transformation is causality-agnostic but necessary for downstream time series analysis tasks, with details introduced in Sec.3.3.

The reasoning for discovering the instantaneous causal effects in the form of the Bayesian Network originates from a widely adopted assumption of causal graph learning (Zheng et al., 2018; Yu et al., 2019; Guo et al., 2021; Geffner et al., 2022; Gong et al., 2023): there exists a ground-truth causal graph $\mathbf{S}^{(t)}$ that specifies instantaneous parents of variables to recover their value generating process. Therefore, in our inner optimization, the goal is to discover the causal structure $\mathbf{S}^{(t)}$ at each time $t$ by recovering the generation of input features $\boldsymbol{H}^{(t)}$. Specifically, given the observed $\boldsymbol{H}^{(t)}$, we aim to estimate a structure $\boldsymbol{A}^{(t)}$, through which a certain distribution $\boldsymbol{Z}^{(t)}$ could generate $\boldsymbol{H}^{(t)}$ for $t \in \{1, \ldots, T\}$. In this way, the instantaneous causal effects are discovered, and the corresponding structures are encoded in $\boldsymbol{A}^{(t)}$. The generation function is expressed as follows.

$$\sum_t log \mathcal{P}(\boldsymbol{H}^{(t)}) = \sum_t log \int \mathcal{P}(\boldsymbol{H}^{(t)}|\boldsymbol{Z}^{(t)}) \mathcal{P}(\boldsymbol{Z}^{(t)}) d\boldsymbol{Z}^{(t)} \tag{3.1}$$

where the generation likelihood $\mathcal{P}(\boldsymbol{H}^{(t)}|\boldsymbol{Z}^{(t)})$ also takes $\boldsymbol{A}^{(t)}$ as input. The complete formula is shown in Eq. 3.3.

For Eq 3.1, on the one hand, it is hard to get the prior distribution $\mathcal{P}(\boldsymbol{Z}^{(t)})$, which is highly related to the distribution of ground-truth causal graph distribution $\mathcal{P}(\mathbf{S}^{(t)})$ at time $t$ (Geffner et al., 2022). On the other hand, for the generation likelihood $\mathcal{P}(\boldsymbol{H}^{(t)}|\boldsymbol{Z}^{(t)})$, the actual posterior $\mathcal{P}(\boldsymbol{Z}^{(t)}|\boldsymbol{H}^{(t)})$ is also intractable. Thus, we resort to the variational autoencoder (VAE) (Kingma and Welling, 2014). In this way, the actual posterior $\mathcal{P}(\boldsymbol{Z}^{(t)}|\boldsymbol{H}^{(t)})$ can be replaced by the variational posterior $\mathcal{Q}(\boldsymbol{Z}^{(t)}|\boldsymbol{H}^{(t)})$, and the prior distribution $\mathcal{P}(\boldsymbol{Z}^{(t)})$ is approximated by a Gaussian distribution. Furthermore, the inside encoder and decoder modules should take the structure $\boldsymbol{A}^{(t)}$ as the input. This design can be realized by various off-the-shelf variational graph autoencoders such as VGAE (Kipf and Welling, 2016), etc. However, the inner optimization is coupled with the outer optimization, i.e., the instantaneous causality will be integrated with cross-time Granger causality to make inferences. The inner complex neural architectures and parameters may render the outer optimization module hard to train, especially when the outer module itself needs to be complex. Therefore, we extend the widely-adopted linear Structural Equation Model (SEM) (Zheng et al., 2018; Yu et al., 2019; Geffner et al., 2022; Gong et al., 2023) to the time-respecting setting as follows.

For $\mathcal{Q}(\boldsymbol{Z}^{(t)}|\boldsymbol{H}^{(t)})$, the encoder equation is expressed as

$$\boldsymbol{Z}^{(t)} = (\boldsymbol{I} - \boldsymbol{A}^{(t)\top}) f_{\theta_{enc}^{(t)}}(\boldsymbol{H}^{(t)}) \tag{3.2}$$

For $\mathcal{P}(\boldsymbol{H}^{(t)}|\boldsymbol{Z}^{(t)})$, the decoder equation is expressed as

$$\boldsymbol{H}^{(t)} = f_{\theta_{dec}^{(t)}}((\boldsymbol{I} - \boldsymbol{A}^{(t)\top})^{-1} \boldsymbol{Z}^{(t)}) \tag{3.3}$$

As analyzed above[2], $f_{\theta_{enc}^{(t)}}$ and $f_{\theta_{dec}^{(t)}}$ do not need complicated neural architectures. Therefore, we can use two-layer MLPs for them. Then, the objective function $\mathcal{L}_{DAG}^{(t)}$ for discovering the instantaneous causality at time $t$ is expressed as follows, which corresponds to the inner optimization.

$$\min_{\theta_{enc}^{(t)}, \theta_{dec}^{(t)}, \boldsymbol{A}^{(t)}} \mathcal{L}_{DAG}^{(t)} = D_{KL}(\mathcal{Q}(\boldsymbol{Z}^{(t)}|\boldsymbol{H}^{(t)}) \| \mathcal{P}(\boldsymbol{Z}^{(t)})) - \mathbb{E}_{\mathcal{Q}(\boldsymbol{Z}^{(t)}|\boldsymbol{H}^{(t)})}[log \mathcal{P}(\boldsymbol{H}^{(t)}|\boldsymbol{Z}^{(t)})]$$

$$\text{s.t. } \sum_t \text{Tr}[(\boldsymbol{I} + \boldsymbol{A}^{(t)} \circ \boldsymbol{A}^{(t)})^N] - N = 0, \text{ for } t \in \{1, \ldots, T\} \tag{3.4}$$

---

[2]The complete forms of $\mathcal{Q}(\boldsymbol{Z}^{(t)}|\boldsymbol{H}^{(t)})$ and $\mathcal{P}(\boldsymbol{H}^{(t)}|\boldsymbol{Z}^{(t)})$ are $\mathcal{Q}_{A^{(t)}}(\boldsymbol{Z}^{(t)}|\boldsymbol{H}^{(t)})$ and $\mathcal{P}_{A^{(t)}}(\boldsymbol{H}^{(t)}|\boldsymbol{Z}^{(t)})$, we omit the subscript $A^{(t)}$ for brevity.

where the first term in $\mathcal{L}_{DAG}^{(t)}$ is the KL-divergence measuring the distance between the distribution of generated $\boldsymbol{Z}^{(t)}$ and the pre-defined Gaussian, and the second term is the reconstruction loss between the generated $\boldsymbol{Z}^{(t)}$ with the original input $\boldsymbol{H}^{(t)}$. Note that there is an important constraint, i.e., $\text{Tr}[(\boldsymbol{I} + \boldsymbol{A}^{(t)} \circ \boldsymbol{A}^{(t)})^N] - N = 0$, on $\boldsymbol{A}^{(t)} \in \mathbb{R}^{N \times N}$. $\text{Tr}(\cdot)$ is the trace of a matrix, and $\circ$ denotes the Hadamard product. The meaning of the constraint is explained as follows. The constraint in Eq. 3.4, i.e., $\text{Tr}[(\boldsymbol{I} + \boldsymbol{A}^{(t)} \circ \boldsymbol{A}^{(t)})^N] - N = 0$ regularizes the acyclicity of $\boldsymbol{A}^{(t)}$ during the optimization process, i.e., the learned $\boldsymbol{A}^{(t)}$ should not have any possible closed-loops at any length.

**Lemma 3.1** *Let $\boldsymbol{A}^{(t)}$ be a weighted adjacency matrix (negative weights allowed). $\boldsymbol{A}^{(t)}$ has no $N$-length loops, if $Tr[(\boldsymbol{I} + \boldsymbol{A}^{(t)} \circ \boldsymbol{A}^{(t)})^N] - N = 0$.*

The intuition is that there will be no $k$-length path from node $u$ to node $v$ on a binary adjacency matrix $\}(u, v) = 0$. Compared with original acyclicity constraints in (Yu et al., 2019), our Lemma 3.1 gets rid of the $\lambda$ condition. Then we can denote $\alpha(A^{(t)}) = \text{Tr}[(\boldsymbol{I} + \boldsymbol{A}^{(t)} \circ \boldsymbol{A}^{(t)})^N] - N$ and use Lagrangian optimization for Eq. 3.4 as follows.

$$\min_{\theta_{enc}^{(t)}, \theta_{dec}^{(t)}, \boldsymbol{A}^{(t)}} \mathcal{L}_{DAG}^{(t)} = D_{KL}(\mathcal{Q}(\boldsymbol{Z}^{(t)}|\boldsymbol{H}^{(t)})\|\mathcal{P}(\boldsymbol{Z}^{(t)})) - \mathbb{E}_{\mathcal{Q}(\boldsymbol{Z}^{(t)}|\boldsymbol{H}^{(t)})}[\log\mathcal{P}(\boldsymbol{H}^{(t)}|\boldsymbol{Z}^{(t)})]$$
$$+ \lambda\,\alpha(A^{(t)}) + \frac{c}{2}|\alpha(A^{(t)})|^2, \quad \text{for } t \in \{1, \ldots, T\} \tag{3.5}$$

where $\lambda$ and $c$ are two hyperparameters, and larger $\lambda$ and $c$ enforce $\alpha(\boldsymbol{A}^{(t)})$ to be smaller.

**Theorem 3.1** *If the ground-truth instantaneous causal graph $\mathbf{S}^{(t)}$ at time $t$ generates the features of variables following the normal distribution, then the inner optimization (i.e., Eq. 3.4) can identify $\mathbf{S}^{(t)}$ under the standard causal discovery assumptions (Geffner et al., 2022).*

## 3.2 OUTER OPTIMIZATION OF TACSAS FOR INTEGRATING INSTANTANEOUS CAUSALITY WITH NEURAL GRANGER CAUSALITY

Given the inner optimization, Bayesian Networks can be obtained at each timestamp $t$, which means that multiple instantaneous causalities are discovered. Thus, in the outer optimization, we integrate these evolving Bayesian Networks into Granger Causality discovery. First, the classic Granger Causality (Granger, 1969) is discovered in the form of the variable-wise coefficients across different timestamps (i.e., a time window) through the autoregressive prediction process. The prediction based on the linear Granger Causality (Granger, 1969) is expressed as follows.

$$\boldsymbol{H}^{(t)} = \sum_{l=1}^{L} \boldsymbol{W}^{(l)} \boldsymbol{H}^{(t-l)} + \boldsymbol{e}^{(t)} \tag{3.6}$$

where $\boldsymbol{H}^{(t)} \in \mathbb{R}^{N \times D}$ denotes the features of $N$ variables at time $t$, $\boldsymbol{e}^{(t)}$ is the noise, and $L$ is the pre-defined time lag indicating how many past timestamps can affect the values of $\boldsymbol{H}^{(t)}$. Weight matrix $\boldsymbol{W}^{(l)} \in \mathbb{R}^{N \times N}$ stores the cross-time coefficients captured by Granger Causality, i.e., matrix $\boldsymbol{W}^{(l)}$ aligns the variables at time $t-l$ with the variables at time $t$. To compute those weights, several linear methods are proposed, e.g., vector autoregressive model (Arnold et al., 2007).

Facing non-linear causal relationships, neural Granger Causality discovery (Tank et al., 2022) is recently proposed to explore the nonlinear Granger Causality effects. The general principle is to represent causal weights $\boldsymbol{W}$ by deep neural networks. To integrate instantaneous effects with neural Granger Causality discovery, our TBN-Granger Causality is defined as follows.

$$\hat{H}(i,:)^{(t)} = f_{\Theta_i}[(\boldsymbol{A}^{(t-1)}, \boldsymbol{H}^{(t-1)}), \ldots, (\boldsymbol{A}^{(t-L)}, \boldsymbol{H}^{(t-L)})] \quad \text{/* TBN-Granger Causality */} \tag{3.7}$$

where $L$ is the lag (or window size) in the Granger Causality, and $i$ is the index of the $i$-th variable. $f_{\Theta_i}$ is a neural computation unit with all parameters denoted as $\Theta_i$, whose input is an $L$-length time-ordered sequence of $(\boldsymbol{A}, \boldsymbol{H})$. And $f_{\Theta_i}$ is responsible for discovering the TBN-Granger Causality for variable $i$ at time $t$ from all variables that occurred in the past time lag $l$. The choice of neural unit $f_{\Theta_i}$ is flexible, such as MLP and LSTM (Tank et al., 2022). Different neural unit choices correspond to different causality interpretations. In our proposed TacSas model, we use graph recurrent neural networks (Wu et al., 2021), and the causality interpretations are introduced in Sec3.3.

In the outer optimization, to evaluate the prediction under the TBN-Granger Causality, we use the mean absolute error (MAE) loss on the prediction and the ground truth, which is effective and widely applied to time-series forecasting tasks (Li et al., 2018; Shang et al., 2021).

$$\min_{\Theta_i, \boldsymbol{A}^{(t-1)}, \ldots, \boldsymbol{A}^{(t-l)}} \mathcal{L}_{pred} = \sum_i \sum_t |H(i, :)^{(t)} - \hat{H}(i, :)^{(t)}| \tag{3.8}$$

where $\Theta_i, \boldsymbol{A}^{(t-1)}, \ldots, \boldsymbol{A}^{(t-l)}$ are all the parameters for the prediction $\hat{H}(i, :)^{(t)}$ of variable $i$ at time $t$. The composition and update rules are expressed below.

**For updating** $f_{\Theta_i}$, we employ the recurrent neural structure to fit the input sequence. Moreover, the sequential inputs also contain the structured data $\boldsymbol{A}$. Therefore, we use the graph recurrent neural architecture (Li et al., 2018) because it is designed for directed graphs, whose core is a gated recurrent unit (Chung et al., 2014).

$$
\begin{aligned}
\boldsymbol{R}^{(t)} &= \text{sigmoid}(\boldsymbol{W}_{\boldsymbol{R}*\boldsymbol{A}^{(t)}}[\boldsymbol{H}^{(t)} \oplus \boldsymbol{S}^{(t-1)}] + \boldsymbol{b}_R) \\
\boldsymbol{C}^{(t)} &= \tanh(\boldsymbol{W}_{\boldsymbol{C}*\boldsymbol{A}^{(t)}}[\boldsymbol{H}^{(t)} \oplus (\boldsymbol{R}^{(t)} \odot \boldsymbol{S}^{(t-1)})] + \boldsymbol{b}_C) \\
\boldsymbol{U}^{(t)} &= \text{sigmoid}(\boldsymbol{W}_{\boldsymbol{U}*\boldsymbol{A}^{(t)}}[\boldsymbol{H}^{(t)} \oplus \boldsymbol{S}^{(t-1)}] + \boldsymbol{b}_U) \\
\boldsymbol{S}^{(t)} &= \boldsymbol{U}^{(t)} \odot \boldsymbol{S}^{(t-1)} + (\boldsymbol{I} - \boldsymbol{U}^{(t)}) \odot \boldsymbol{C}^{(t)}
\end{aligned}
\tag{3.9}
$$

where $\boldsymbol{R}^{(t)}$, $\boldsymbol{C}^{(t)}$, and $\boldsymbol{U}^{(t)}$ are three parameterized gates, with corresponding weights $\boldsymbol{W}$ and bias $\boldsymbol{b}$. $\boldsymbol{H}^{(t)}$ is the input, and $\boldsymbol{S}^{(t)}$ is the hidden state. Gates $\boldsymbol{R}^{(t)}$, $\boldsymbol{C}^{(t)}$, and $\boldsymbol{U}^{(t)}$ share the similar structures. For example, in $\boldsymbol{R}^{(t)}$, the graph convolution operation for computing the weight $\boldsymbol{W}_{\boldsymbol{R}*\boldsymbol{A}^{(t)}}$ is defined as follows, and the same computation applies to gates $\boldsymbol{U}^{(t)}$ and $\boldsymbol{C}^{(t)}$.

$$\boldsymbol{W}_{\boldsymbol{R}*\boldsymbol{A}^{(t)}} = \sum_{k=0}^{K} \theta_{k,1}^R (\boldsymbol{D}_{out}^{(t)}{}^{-1} \boldsymbol{A}^{(t)})^k + \theta_{k,2}^R (\boldsymbol{D}_{in}^{(t)}{}^{-1} \boldsymbol{A}^{(t)}{}^{\top})^k \tag{3.10}$$

where $\theta_{k,1}^R$, $\theta_{k,2}^R$ are learnable weight parameters; scalar $k$ is the order for the stochastic diffusion operation (i.e., similar to steps of random walks); $\boldsymbol{D}_{out}^{(t)}{}^{-1} \boldsymbol{A}^{(t)}$ and $\boldsymbol{D}_{in}^{(t)}{}^{-1} \boldsymbol{A}^{(t)}{}^{\top}$ serve as the transition matrices with the in-degree matrix $\boldsymbol{D}_{in}^{(t)}$ and the out-degree matrix $\boldsymbol{D}_{out}^{(t)}$; $-1$ and $\top$ are inverse and transpose operations.

**For updating each of** $\{\boldsymbol{A}^{(t-1)}, \ldots, \boldsymbol{A}^{(t-l)}\}$, we take $\boldsymbol{A}^{(t-l)}$ as an example to illustrate. The optimal $\boldsymbol{A}^{(t-l)}$ stays in the space of $\{0, 1\}^{N \times N}$. To be specific, each edge $A^{(t-l)}(i, j)$ can be parameterized as $\theta_{i,j}^{(t-l)}$ following the Bernoulli distribution. However, $N^2 l$ is hard to scale, and the discrete variables are not differentiable. Therefore, we adopt the Gumbel reparameterization from (Jang et al., 2017; Maddison et al., 2017). It provides a continuous approximation for the discrete distribution, which has been widely used in the graph structure learning (Kipf et al., 2018; Shang et al., 2021). The general reparameterization form can be written as $A^{(t-l)}(i, j) = softmax(FC((H(i, :)^{(t-l)} \| H(j, :)^{(t-l)}) + g)/\xi)$, where $FC$ is a feedforward neural network, $g$ is a scalar drawn from a Gumbel$(0, 1)$ distribution, and $\xi$ is a scaling hyperparameter. Different from (Kipf et al., 2018; Shang et al., 2021), in our setting, the initial structure input is constrained by the causality discovery, which originates from the inner optimization step. Hence, the structure learning in the outer optimization takes the adjacency matrix from the inner optimization as the initial input, which is

$$A_{outer}^{(t-l)}(i, j) = softmax(A_{inner}^{(t-l)}(i, j) + \boldsymbol{g})/\xi) \tag{3.11}$$

where $A_{inner}^{(t)}(i, j)$ is the structure learned by our inner optimization through Eq. 3.4, $A_{outer}^{(t)}(i, j)$ is the updated structure, and $\boldsymbol{g}$ is a vector of i.i.d samples drawn from a Gumbel$(0, 1)$ distribution. In outer optimization, Eq. 3.8 fine-tunes the evolving Bayesian Networks to make the intra-time causality fit the cross-time causality well. Note that, the outer optimization w.r.t. $\boldsymbol{A}^{(t)}$ may break the acyclicity, and another round of inner optimization may be necessary.

### 3.3 DEPLOYMENT OF TACSAS FOR TIME SERIES FORECASTING AND ANOMALY DETECTION

In this section, we introduce how TacSas achieves tensor time series forecasting and anomaly detection in threefold: data preprocessing, neural architecture selection, and training procedure.

**First (data preprocessing)**, in addition to forecasting, TacSas is also for anomaly detection. Thus, we design the hidden feature $\mathcal{H}$ extraction in TacSas motivated by the Extreme Value Theory (Beirlant et al., 2004) or so-called Extreme Value Distribution (Siffer et al., 2017) in streaming data.

**Remark 3.1** *According to the Extreme Value Distribution (Fisher and Tippett, 1928), under the limiting forms of frequency distributions, extreme values have the same kind of distribution, regardless of original distributions.*

An example (Siffer et al., 2017) can help interpret and understand the Extreme Value Distribution theory. Maximum temperatures or tide heights have more or less the same distribution even though the distributions of temperatures and tide heights are not likely to be the same. As rare events have a lower probability, there are only a few possible shapes for a general distribution to fit. Inspired by this observation, we can design a simple but effective module in TacSas to achieve anomaly detection, i.e., a pre-trained autoencoder model that tries to explore the distribution of normal features in $\mathcal{X}$ as shown in Figure 2. As long as this autoencoder model can capture the latent distribution for normal events, then the generation probability of a piece of time series data can be utilized as the condition for detecting anomaly patterns. This is because the extreme values are identified with a remarkably low generation probability. To be specific, after the forecast $\boldsymbol{H}^{(t)}$ is output, the generation probability of $\boldsymbol{H}^{(t)}$ into $\boldsymbol{X}^{(t)}$ through the pre-trained autoencoder can be used to detect the anomalies at $t$.

**Second (neural architecture selection)**, we encode $f_{\Theta_i}$ into a sequence-to-sequence model (Sutskever et al., 2014). That is, given a time window (or time lag), TacSas could forecast the corresponding features for the next time window. Moreover, with $\boldsymbol{W}^{(l)}$ in Eq. 3.6 and $f_{\Theta_i}$ in Eq. 3.7, we can observe that the classical linear Granger Causality $\boldsymbol{W}^{(l)}$ can be discovered for each time lag. In other words, each time lag has its own discovered coefficients, but $f_{\Theta_i}$ is shared by all time lags. This sharing manner is designed for scalability and is called Summary Causal Graph (Marcinkevics and Vogt, 2021; Assaad et al., 2022). The underlying intuition is that the causal effects mainly depend on the near timestamps. Further, for the neural Granger Causality interpretation in $f_{\Theta_i}$, we follow the rule (Tank et al., 2022) that if the $j$-th row of ($\boldsymbol{W}_{\boldsymbol{R}*\boldsymbol{A}^{(t)}}$, $\boldsymbol{W}_{\boldsymbol{C}*\boldsymbol{A}^{(t)}}$, and $\boldsymbol{W}_{\boldsymbol{U}*\boldsymbol{A}^{(t)}}$) are zeros, then variable $j$ is not the Granger-cause for variable $i$ in this time window.

**Third (training procedure)**, as shown in Figure 2, the autoencoder can be pre-trained with reconstruction loss (e.g., MSE) ahead of the inner and outer optimization, to obtain $\mathcal{H}$ for the feature latent distribution representation. By utilizing all input $\mathcal{H}$, the inner optimization learns the sequential Bayesian Networks, and the outer optimization aligns Bayesian Networks with the neural Granger Causality to produce all the forecast $\mathcal{H}'$. The inner and outer optimization can be trained interchangeably.

## 4 EXPERIMENTS

The ground-truth causality discovery experiments in the synthetic benchmark, Lorenz 96 System (Lorenz, 1996), are shown in Appendix B.1, where our TacSas can capture the true causality with the competitive high accuracy. Then, in this section, we test TacSas on utilizing its discovery for time series forecasting and anomaly detection.

### 4.1 EXPERIMENT SETUP

**Datasets**. Our forecasting data (i.e., hourly tensor time series data) originates from climate domain benchmark ERA5 (Hersbach et al., 2018)[3]. To be specific, we select four datasets covering 45 weather features (i.e., wind gusts, rain, etc.) from 238[4] counties in the United States of America during 2017–2020. Moreover, we choose thunderstorms as the anomaly pattern to be detected after forecasting. The thunderstorm record is identified in NOAA database[5] hourly and nationwide, i.e., 1 means a thunderstorm happens in the corresponding hour at a certain location, and 0 means no thunderstorm happens. We processed the geocode to align weather features in ERA5 with anomaly patterns in NOAA. The geographic distribution and anomaly pattern frequency distribution are shown in Appendix E.

---

[3]https://cds.climate.copernicus.eu/cdsapp#!/home

[4]100 of 238 counties are top-ranked counties for the thunderstorm (anomaly label) frequency, and the rest are randomly selected.

[5]https://www.ncdc.noaa.gov/stormevents/ftp.jsp

**Baselines**. Besides the causality discovery baseline in Appendix B, the **first** category is for tensor time series forecasting: (1) GRU (Chung et al., 2014) is a classical sequence to sequence generation model. (2) DCRNN (Li et al., 2018) is a graph convolutional recurrent neural network, of which the input graph structure is given, not causal, and static (i.e., shared by all timestamps). In this viewpoint, we let each node randomly distribute its unit weights to others. (3) GTS (Shang et al., 2021) is also a graph convolutional recurrent neural network that does not need the input graph but learns the structure based on the node features, but the learned structure is also shared by all timestamps and is not causal. To compare the performance of DCGNN (Li et al., 2018) and GTS (Shang et al., 2021) with TacSas, causality is the control variable since we make all the rest (e.g., neural network type, number of layers, etc.) identical for them. The **second** category is for anomaly detection on tensor time series: (1) DeepSAD (Ruff et al., 2020), (2) DeepSVDD (Ruff et al., 2018), and (3) DROCC (Goyal et al., 2020). Since these three have no forecast abilities, we let them use the ground-truth observations, and our TacSas utilizes the forecast features during anomaly detection experiments. Also, these three baselines are designed for multi-variate time-series data, not tensor time-series. Thus, we flatten our tensor time series along the spatial dimension and report the average performance for these three baselines over all locations.

Next, we introduce forecasting and anomaly detection performance. Details about split and hyper-parameters are in Appendix C. More ablation studies can be found in Appendix B.3.

## 4.2 FORECASTING PERFORMANCE

In Table 1, we present the forecasting performance in terms of mean absolute error (MAE) on the testing data of three algorithms, namely DCGNN (Li et al., 2018), GTS (Shang et al., 2021), ST-SSL (Ji et al., 2023), our TacSas, and TacSas++ (i.e., TacSas with persistence forecast constraints). Here, we set the time window as 24, meaning that we use the past 24 hours tensor time series to forecast the future 24 hours in an autoregressive manner. Moreover, for baselines and TacSas, we set $f_{\Theta_i}$ in Eq.3.7 shared by all weather variables to ensure the scalability, such that we do not need to train $N$ recurrent graph neural networks for a single prediction. In Table 1, we can observe a general pattern that our TacSas outperforms the baselines with GTS performing better than DCGNN. For example, with 2017 as the testing data, our TacSas performs 39.44% and 36.16% better than DCRNN and GTS. An explanation is that the temporally fine-grained causal relationships can contribute more to the forecasting accuracy than non-causal directed graphs, since DCGNN, GTS, and our TacSas all share the graph recurrent manner. TacSas however, discovers causalities at different timestamps, while DCGNN and GTS use feature similarity based connections. Moreover, ST-SSL achieves competitive forecasting performance via contrastive learning on time series. Motivated by contrastive manner, TacSas++ is proposed by persistence forecast constraints. That is, the current forecast of TacSas is further calibrated by its nearest time window (i.e., the last 24 hours in our setting).

Table 1: Forecasting Error (MAE, $10^{-2}$)

|  | ERA5-2017 ($\downarrow$) | ERA5-2018 ($\downarrow$) | ERA5-2019 ($\downarrow$) | ERA5-2020 ($\downarrow$) |
|---|---|---|---|---|
| GRU | $1.8834 \pm 0.0126$ | $1.9764 \pm 0.1466$ | $1.6194 \pm 0.2645$ | $1.7859 \pm 0.2324$ |
| DCRNN | $0.0819 \pm 0.0025$ | $0.0797 \pm 0.0049$ | $0.0799 \pm 0.0035$ | $0.0826 \pm 0.0033$ |
| GTS | $0.0777 \pm 0.0054$ | $0.0766 \pm 0.0029$ | $0.0760 \pm 0.0031$ | $0.0742 \pm 0.0021$ |
| TacSas | $0.0496 \pm 0.0017$ | $0.0499 \pm 0.0017$ | $0.0502 \pm 0.0016$ | $0.0488 \pm 0.0019$ |
| ST-SSL | $0.0345 \pm 0.0051$ | $0.0330 \pm 0.0018$ | $0.0361 \pm 0.0021$ | $0.0348 \pm 0.0020$ |
| TacSas++ | $\mathbf{0.0271 \pm 0.0004}$ | $\mathbf{0.0276 \pm 0.0004}$ | $\mathbf{0.0282 \pm 0.0003}$ | $\mathbf{0.0265 \pm 0.0004}$ |

To evaluate our explanation, we visualize causal connections at different times in Figure 3. Specifically, we show the Bayesian Network of 238 counties at the same hour on two consecutive days in the training data (i.e., May 1st and May 2nd, 2018). Interestingly, we can observe that two patterns in Figure 3 are almost identical at first glance. That could be the reason why DCRNN and GTS can perform well using the static structure. However, upon closer inspection, we find that these two are quite dif-

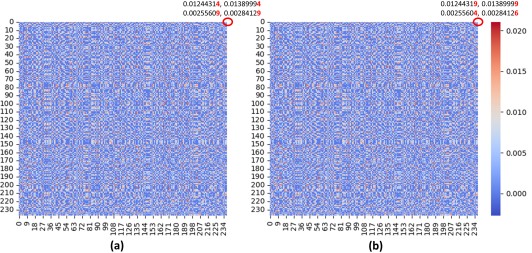

Figure 3: Time-Respecting Bayesian Networks of at the Same Hour of Two Consecutive Days.

ferent to some extent if we zoom in, such as, in the upper right corner. Although the values have a tiny divergence, their volume is quite large. In two matrices of Figure 3, the number of different cells is 28,509, and the corresponding percentage is $\frac{28509}{238 \times 238} \approx 0.5033$. We suppose that discovering those value-tiny but volume-big differences makes TacSas outperform, to a large extent.

## 4.3 ANOMALY DETECTION

After forecasting, we can have the hourly forecast of weather features at certain locations, denoted as $\mathcal{X}'$. Then, we use the encoder-decoder model in Figure 2 to calculate the feature-wise generation probability using mean squared error (MSE) between $\mathcal{X}'$ and its generation $\bar{\mathcal{X}}'$. Thus, we can calculate the average of feature-wise generation probability as the condition of anomalies to identify if an anomaly weather pattern (e.g., a thunderstorm) happens in an hour in a particular location. In Table 2, we use the Area Under the ROC Curve (i.e., AUC-ROC) as the metric, repeat the experiments four times, and report the performance of TacSas with baselines.

Table 2: Anomaly Detection Performance (AUC-ROC)

|  | NOAA-2017 ($\uparrow$) | NOAA-2018 ($\uparrow$) | NOAA-2019 ($\uparrow$) | NOAA-2020 ($\uparrow$) |
|---|---|---|---|---|
| DeepSAD | $0.5305 \pm 0.0481$ | $0.5267 \pm 0.0406$ | $0.5563 \pm 0.0460$ | $0.6420 \pm 0.0054$ |
| DeepSVDD | $0.5201 \pm 0.0045$ | $0.5603 \pm 0.0111$ | $\mathbf{0.6784 \pm 0.0112}$ | $0.5820 \pm 0.0205$ |
| DROCC | $0.5319 \pm 0.0661$ | $0.5103 \pm 0.0147$ | $0.6236 \pm 0.0992$ | $0.5630 \pm 0.1082$ |
| TacSas | $\mathbf{0.5556 \pm 0.0010}$ | $\mathbf{0.5685 \pm 0.0011}$ | $0.6298 \pm 0.0184$ | $\mathbf{0.6745 \pm 0.0185}$ |

From Table 2, we can observe that the detection module of TacSas achieves very competitive performance. An explanation is that, based on the anomalies distribution shown in Table 3, it can be observed that the anomalies are very rare events. Our generative manner could deal with the very rare scenario by learning the feature latent distributions instead of the (semi-)supervised learning manner. For example, the maximum frequency of occurrences of thunderstorms is 770 (i.e., Jun 2017), which is collected from 238 counties over $30 \times 24 = 720$ hours, and the corresponding percentage is $\frac{770}{238 \times 30 \times 24} \approx 0.45\%$. Recall Remark 3.1, facing such rare events, we possibly find a single distribution to fit various anomaly patterns.

## 5 RELATED WORK

In recent times, there has been a growing focus on structured learning in the context of time series data (Li et al., 2018; Wu et al., 2020; Zhao et al., 2020a; Cao et al., 2020; Shang et al., 2021; Deng and Hooi, 2021; Marcinkevics and Vogt, 2021; Geffner et al., 2022; Tank et al., 2022; Spadon et al., 2022; Gong et al., 2023), which learned structures contribute to various time series analysis tasks like forecasting, anomaly detection, imputation, etc. As a directed and interpretable structure, causal graphs attract much research attention in this research topic (Guo et al., 2021). Granger Causality is a classic tool for discovering the cross-time variable causality in time series (Granger, 1969; Arnold et al., 2007). Facing complex patterns in time series data, different upgraded Granger Causality discovery methods emerge in different directions. Also, neural Granger Causality tools are recently proposed (Tank et al., 2022; Nauta et al., 2019; Khanna and Tan, 2020; Marcinkevics and Vogt, 2021; Xu et al., 2019; Huang et al., 2020), which utilizes the deep neural network to discover the nonlinear Granger causal coefficients and serve for the time-series forecasting tasks better. For example, in (Tank et al., 2022), authors introduce how to use multi-layer perception (MLPs) and long short-term memory (LSTMs) to realize the Neural Granger Causality for the forecasting task and how to interpret the Granger causal coefficients from neurons in deep networks. However, Granger Causality or Neural Granger Causality focuses on cross-time variable causality discovery and overlooks the instantaneous (or intra-time) variable causality. Also, how to utilize the discovered comprehensive causality to contribute to the downstream time series analysis tasks is under-explored mainly, especially in a setting where the ground-truth causal structures are hardly available for evaluation. More related work for spatial-temporal data forecasting and anomaly detection is discussed in Appendix D.

## 6 CONCLUSION

In this paper, we first propose TBN-Granger Causality to align the instantaneous causal effects with time-lagged Granger causality. Moreover, we design TacSas to use TBN-Granger Causality on time series analysis tasks like forecasting and anomaly detection in the real-world tensor time-series data and perform extensive experiments, where the results show the effectiveness of TacSas.

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

# A THEORETICAL ANALYSIS

## A.1 PROOF OF LEMMA 3.1

Following (Yu et al., 2019), at each time $t$, we can extend $(\boldsymbol{I} + \boldsymbol{A}^{(t)} \circ \boldsymbol{A}^{(t)})^N$ by binomial expansion as follows.

$$(\boldsymbol{I} + \boldsymbol{A}^{(t)} \circ \boldsymbol{A}^{(t)})^N = \boldsymbol{I} + \sum_{k=1}^{N} \binom{N}{k} (\boldsymbol{A}^{(t)})^k \tag{A.1}$$

Since

$$\boldsymbol{I} \in \mathbb{R}^{N \times N} \tag{A.2}$$

then

$$\mathrm{Tr}(\boldsymbol{I}) = N \tag{A.3}$$

Thus, if

$$(\boldsymbol{I} + \boldsymbol{A}^{(t)} \circ \boldsymbol{A}^{(t)})^N - N = 0 \tag{A.4}$$

then

$$(\boldsymbol{A}^{(t)})^k = 0, \text{ for any } k \tag{A.5}$$

Therefore, $\boldsymbol{A}^{(t)}$ is acyclic, i.e., no closed-loop exists in $\boldsymbol{A}^{(t)}$ at any possible length. Overall, the general idea of Lemma 3.1 is to ensure that the diagonal entries of the powered adjacency matrix have no 1s. There are also other forms for acyclicity constraints obeying the same idea but in different expressions, like exponential power form in (Zheng et al., 2018).

## A.2 SKETCH PROOF OF THEOREM 3.1

According to Theorem 1 from (Geffner et al., 2022), the ELBO form as our Eq. 3.4 could identity the ground-truth causal structure $\mathbf{S}^{(t)}$ at each time $t$. The difference between our ELBO and the ELBO in (Geffner et al., 2022) is entries in the KL-divergence. Specifically, in (Geffner et al., 2022), the prior and variational posterior distributions are on the graph level. Usually, the prior distribution of graph structures is not easy to obtain (e.g., the non-IID and heterophyllous properties). Then, we transfer the graph structure distribution to the feature distribution that the Gaussian distribution can model. That's why our prior and variational posterior distributions in the KL-divergence are on the feature (generated by the graph) level.

# B EMPIRICAL ANALYSIS

## B.1 GROUND-TRUTH CAUSALITY DISCOVERY ABILITY OF TACSAS

Lorenz-96 model (Lorenz, 1996) is a famous synthetic system of multivariate time-series, e.g., $\boldsymbol{X} \in \mathbb{R}^{P \times T}$ is a $P$-dimensional time series whose dynamics can be modeled as follows.

$$\frac{d\boldsymbol{X}(i,t)}{dt} = (\boldsymbol{X}(i+1,t) - \boldsymbol{X}(i-2,t))\boldsymbol{X}(i-1,t) - \boldsymbol{X}(i,t) + F, \text{ for } i \in \{1, 2, \ldots, P\} \tag{B.1}$$

where $\boldsymbol{X}(0,t) = \boldsymbol{X}(P,t)$, $\boldsymbol{X}(-1,t) = \boldsymbol{X}(P-1,t)$, $\boldsymbol{X}(P+1,t) = \boldsymbol{X}(1,t)$, and $F$ is the forcing constant determining the level of nonlinearity and chaos in the time series. With the above modeling, the corresponding ground-truth Granger causal structures can be simulated, involving multivariate, nonlinear, and sparse (Tank et al., 2022).

To generate the ground-truth causal structures, there are two parameters, i.e., the number of variables (i.e., $P$) and the number of timestamps (i.e., $T$). Therefore, we control these two parameters and report the accuracy of TacSas discovered causal structures against the ground-truth ones (i.e., 0/1 adjacency matrices), compared with the state-of-the-art causality discovery method GVAR (Marcinkevics and Vogt, 2021). The comparison is shown in Figure 4 after eight experiment trials with mean and variance computed, where we can observe our TacSas achieve the competitive accuracy of discovering the ground-truth causal structures. Also, by comparing Figure 4(a) and (b) (and Figure 4(c)

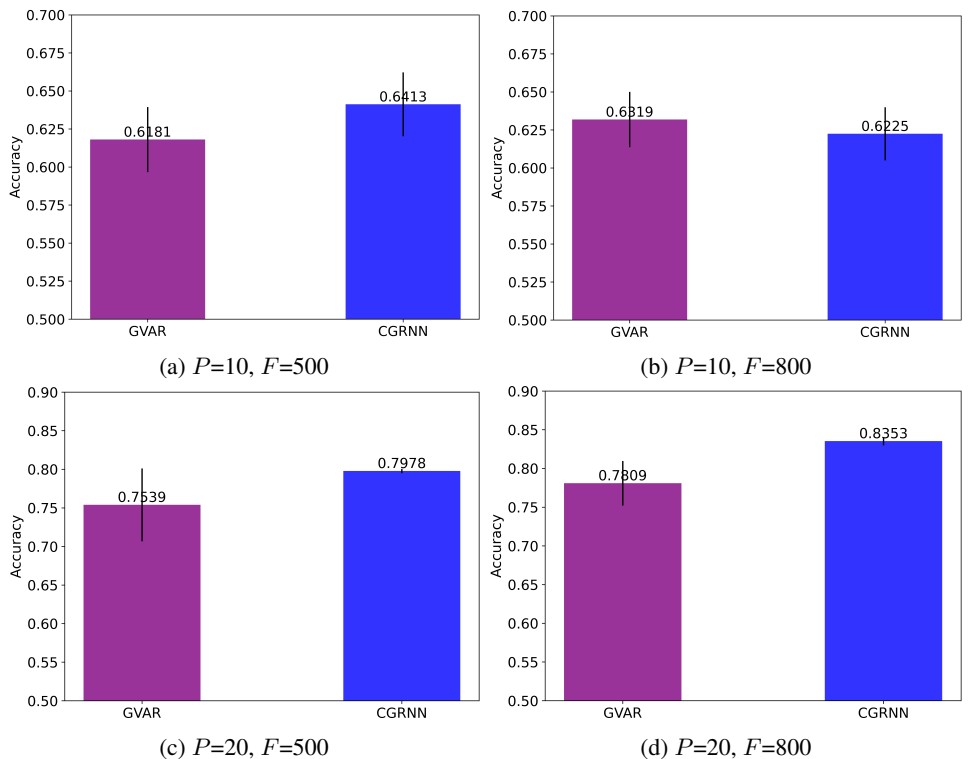

Figure 4: Accuracy of Causality Discovery in Lorenz-96 with Varying Number of Variables (i.e., $P$) and Timestamps (i.e., $T$).

and (d)), we can see that fixing the number of variables (i.e., $P$), increasing the time series length (i.e., $T$) may help discover the causality. And by comparing Figure 4(a) and (c) (and Figure 4(b) and (d)), we can see that fixing the time length (i.e., $T$), increasing the number of variables (i.e., $P$) may make the causality easier to be discovered.

### B.2 VALIDATION OF ANOMALY DETECTION ABILITY OF TACSAS

Another capability of TacSas is anomaly detection. Based on the analysis of Remark 3.1, the detection function of TacSas originates from the accurate expression of the feature distribution. Although our forecast features have better accuracy than selected baselines (e.g., DCGNN and GTS), we need to verify if the forecast features still have a negligible divergence from the ground-truth features in terms of distribution. If so, we can safely use the forecast features to detect anomalies.

Therefore, we design the ablation study. We remove the forecasting part of TacSas i.e., we let the encoder and decoder in Figure 2 directly learn the distribution of ground-truth features (instead of forecast features) and then test reconstruction loss on ground-truth features. In Figure 5, we show the feature reconstruction loss (i.e., mean squared error) curve of the encoder and decoder on the validation set as the epoch increases. After the training of the encoder and decoder is converged, we can also observe that the ground-truth feature reconstruction loss does not have a very large divergence from the forecast features. Now, we are ready to do the following anomaly detection experiments.

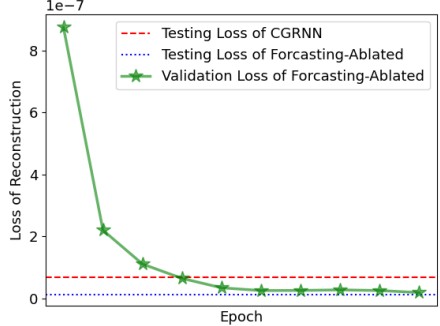

Figure 5: Ablation of TacSas on Cross-Validation Group #2 (i.e., 2018 as testing)

### B.3 ABLATION STUDY

As shown in Table 1, the GRU (Chung et al., 2014) method does not perform well. A latent reason is that it can not take any structural information from the time series. Motivated by this guess, we designed the following ablation study on the forecasting task. The ablated TacSas is designed by only keeping the forget gate in Eq. 3.9, i.e., the last equation in Eq. 3.9, then all the rest of the gates follow the GRU method. As shown in Figure 6, we can see that only taking partial time-respecting causal structural information could not enable TacSas to achieve the best performance, but accepting this partial information can help GRU improve the performance compared with Table 1.

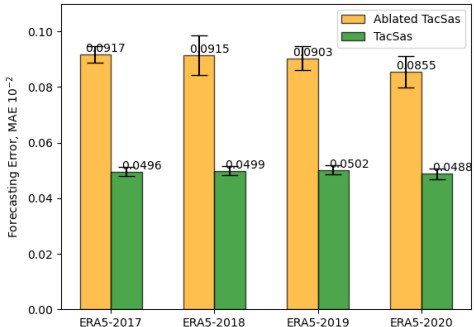

Figure 6: Ablation Study on Forecasting Task.

## C IMPLEMENTATION

### C.1 HYPERPARAMETER SEARCH

In Eq. 3.5, instead of fixing the hyperparameter $\lambda$ and $c$ during the optimization. Increasing the values of hyperparameter $\lambda$ and $c$ can reduce the possibility that learned structures break the acyclicity (Yu et al., 2019), such that one iterative way to increase hyperparameters $\lambda$ and $c$ during the optimization can be expressed as follows.

$$\lambda_{i+1} \leftarrow \lambda_i + c_i \alpha(\boldsymbol{A}_i^{(t)}) \tag{C.1}$$

and

$$c_{i+1} = \begin{cases} \eta c_i & \text{if } |\alpha(\boldsymbol{A}_i^{(t)})| > \gamma |\alpha(\boldsymbol{A}_{i-1}^{(t)})| \\ c_i & \text{otherwise} \end{cases} \tag{C.2}$$

where $\eta > 1$ and $0 < \gamma < 1$ are two hyperparameters, the condition $|\alpha(\boldsymbol{A}_i^{(t)})| > \gamma |\alpha(\boldsymbol{A}_{i-1}^{(t)})|$ means that the current acyclicity $\alpha(\boldsymbol{A}_i^{(t)})$ at the $i$-th iteration is not ideal, because it is not decreased below the $\gamma$ portion of $\alpha(\boldsymbol{A}_{i-1}^{(t)})$ from the last iteration $i-1$.

### C.2 REPRODUCIBILITY

For forecasting and anomaly detection, we have four cross-validation groups. For example, focusing on an interesting time interval each year (e.g., from May to August is the season for frequent thunderstorms), we set group #1 with [2018, 2019, 2020] as training, [2021] as validation, and [2017] as testing. Thus, we have 8856 hours, 45 weather features, and 238 counties in the training set. The rest three groups are {[2019, 2020, 2021], [2017], [2018]}, {[2020, 2021, 2017], [2018], [2019]}, and {[2021, 2017, 2018], [2019], [2020]}, respectively. Therefore, TacSas and baselines are required to forecast the testing set and detect the anomaly patterns in the testing set.

The synthetic data is publicly available [6]. According to the corporate policy, our contributed data and the code of TacSas will be released after the paper is published. The experiments are programmed based on Python and Pytorch on a Windows machine with 64GB RAM and a 16GB RTX 5000 GPU.

---

[6]https://github.com/i6092467/GVAR

## D    FURTHER DISCUSSION FOR SPATIAL-TEMPORAL DATA FORECASTING AND ANOMALY DETECTION

In addition to the related work presented in Section 5, our discussion delves further into forecasting and anomaly detection methodologies specifically applied to time series analysis. Noteworthy applications of graph learning techniques in time series forecasting span diverse domains, including but not limited to heatwave prediction (Li et al., 2023), frost forecasts (Lira et al., 2022), identification of underlying fields in charged particle environments (Kofinas et al., 2021), and traffic forecasting (Yu et al., 2018; Song et al., 2020; Li and Zhu, 2021). To elucidate, our focus narrows down to modeling temporal interactions among objects from the perspective of latent fields. Specifically, authors proposed an equivariant graph network in (Kofinas et al., 2023). This innovative approach integrates field forces to unveil underlying fields within intricate spatial-temporal contexts. Moreover, a spatial self-supervised learning paradigm is introduced in the study by (Ji et al., 2023). This paradigm comprises an adaptive graph augmentation and a clustering-based generative task. Additionally, a temporal self-supervised learning paradigm relies on a time-aware contrastive task, augmenting the primary task of traffic flow prediction with crucial spatial and temporal signals. Beyond the realm of value anomaly detection, our exploration extends to structural anomaly detection, exemplified by works such as (Zhang et al., 2019). This particular approach is designed to discern latent relational anomalies within graph structures. Confronted with diverse anomaly patterns in time series, the proposition of a universal anomaly detection method capable of addressing all conceivable anomaly patterns emerges as a compelling avenue for future research, prompting our interest in further exploration in this direction.

## E    NEW TENSOR TIME SERIES DATASET

In this section, we introduce the details of our contributed dataset in terms of meaning, statistics, and distributions of features and anomaly labels.

### E.1    GEOGRAPHIC DISTRIBUTION OF THE TIME SERIES DATA

The geographic distribution of 238 selected counties in the United States of America is shown in Figure 7, where the circle with numbers denotes the aggregation of spatially near counties. Of 238 selected counties, 100 are selected for the top-ranked counties based on the yearly frequency of thunderstorms. The rest are selected randomly and try to provide extra information (e.g., causality discovery).

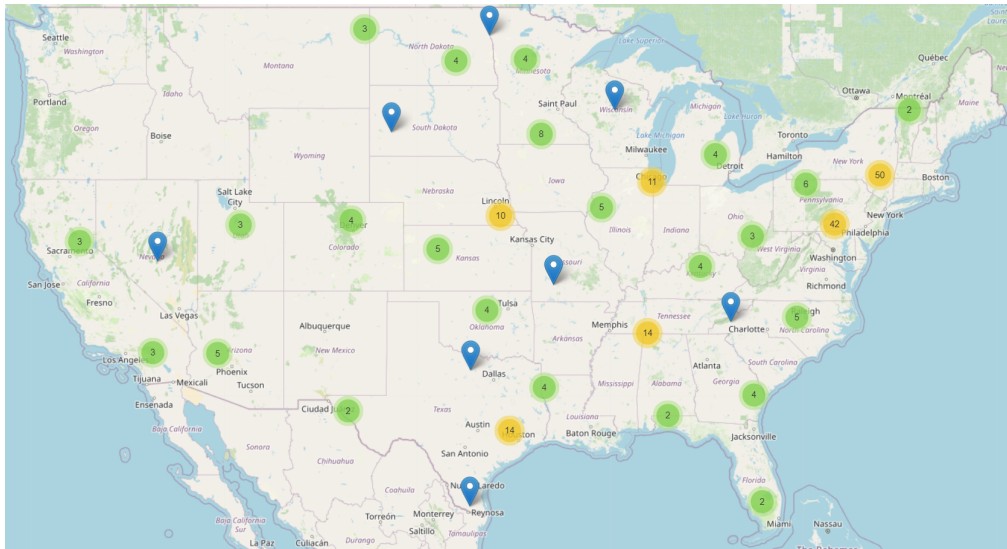

Figure 7: Geographic Distribution of Covered Counties in the Time Series Dataset (The number in the circle stands for the aggregation of nearby counties).

E.2 ABNORMAL PATTERNS OF THE TIME SERIES DATA

Table 3: Statistics of Anomaly Weather Patterns (i.e., Thunderstorm Wind) Occurrence in 238 Selected Counties in the United States

| Year | 2017 | 2018 | 2019 | 2020 | 2021 |
|------|------|------|------|------|------|
| Jan | 26 | 3 | 2 | 41 | 7 |
| Feb | 53 | 6 | 9 | 50 | 8 |
| Mar | 85 | 16 | 26 | 63 | 62 |
| Apr | 93 | 44 | 140 | 170 | 60 |
| May | 245 | 207 | 263 | 175 | 218 |
| Jun | 770 | 302 | 348 | 331 | 452 |
| Jul | 306 | 291 | 457 | 453 | 701 |
| Aug | 294 | 269 | 415 | 354 | 435 |
| Sep | 61 | 80 | 122 | 29 | 123 |
| Oct | 32 | 32 | 82 | 60 | 55 |
| Nov | 20 | 22 | 9 | 114 | 11 |
| Dec | 5 | 15 | 11 | 8 | 58 |

E.3 FEATURE DESCRIPTION OF THE TIME SERIES DATA

Table 4: Feature Descriptions with Instance Values Sampled from Jefferson, Alabama U.S. on 9:00-10:00, 01/05/2017, UTC

| Feature | Unit | Description | Value |
|---------|------|-------------|-------|
| 100-meter wind towards east | m s$^{-1}$ | This parameter is the eastward component of the 100 m wind. It is the horizontal speed of air moving towards the east, at a height of 100 meters above the surface of the Earth, in meters per second. Care should be taken when comparing model parameters with observations, because observations are often local to a particular point in space and time, rather than representing averages over a model grid box. This parameter can be combined with the northward component to give the speed and direction of the horizontal 100 m wind. | -3.192476 |
| 100-meter wind towards north | m s$^{-1}$ | This parameter is the northward component of the 100 m wind. It is the horizontal speed of air moving towards the north, at a height of 100 meters above the surface of the Earth, in meters per second. Care should be taken when comparing model parameters with observations, because observations are often local to a particular point in space and time, rather than representing averages over a model grid box. This parameter can be combined with the eastward component to give the speed and direction of the horizontal 100 m wind. | -1.892055 |

| 10-meter wind gust (maximum) | m s$^{-1}$ | Maximum 3-second wind at 10 m height as defined by WMO. Parametrization represents turbulence only before 01102008; thereafter effects of convection are included. The 3 s gust is computed every time step, and the maximum is kept since the last postprocessing. | 3.620435 |
|---|---|---|---|
| 10-meter wind gust (instantaneous) | m s$^{-1}$ | This parameter is the maximum wind gust at the specified time, at a height of ten meters above the surface of the Earth. The WMO defines a wind gust as the maximum of the wind averaged over 3-second intervals. This duration is shorter than a model time step, and so the ECMWF Integrated Forecasting System (IFS) deduces the magnitude of a gust within each time step from the time-step-averaged surface stress, surface friction, wind shear, and stability. Care should be taken when comparing model parameters with observations, because observations are often local to a particular point in space and time, rather than representing averages over a model grid box. | 3.178461 |
| 10-meter wind towards east | m s$^{-1}$ | This parameter is the eastward component of the 10m wind. It is the horizontal speed of air moving towards the east, at a height of ten meters above the surface of the Earth, in meters per second. Care should be taken when comparing this parameter with observations because wind observations vary on small space and time scales and are affected by the local terrain, vegetation, and buildings that are represented only on average in the ECMWF Integrated Forecasting System (IFS). This parameter can be combined with the V component of 10m wind to give the speed and direction of the horizontal 10m wind. | -1.094084 |
| 10-meter wind towards north | m s$^{-1}$ | This parameter is the northward component of the 10m wind. It is the horizontal speed of air moving towards the north, at a height of ten metres above the surface of the Earth, in metres per second. Care should be taken when comparing this parameter with observations, because wind observations vary on small space and time scales and are affected by the local terrain, vegetation and buildings that are represented only on average in the ECMWF Integrated Forecasting System (IFS). This parameter can be combined with the U component of 10m wind to give the speed and direction of the horizontal 10m wind. | -1.119224 |

| Atmospheric water content | kg m$^{-2}$ | This parameter is the sum of water vapor, liquid water, cloud ice, rain, and snow in a column extending from the surface of the Earth to the top of the atmosphere. In old versions of the ECMWF model (IFS), rain and snow were not accounted for. | 9.287734 |
|---|---|---|---|
| Atmospheric water vapor content | kg m$^{-2}$ | This parameter is the total amount of water vapor in a column extending from the surface of the Earth to the top of the atmosphere. This parameter represents the area averaged value for a grid box. | 9.287452 |
| Dewpoint | K | This parameter is the temperature to which the air, at 2 meters above the surface of the Earth, would have to be cooled for saturation to occur. It is a measure of the humidity of the air. Combined with temperature and pressure, it can be used to calculate relative humidity. 2m dew point temperature is calculated by interpolating between the lowest model level and the Earth's surface, taking account of the atmospheric conditions. This parameter has units of kelvin (K). Temperature measured in kelvin can be converted to degrees Celsius (°C) by subtracting 273.15. | 269.059570 |
| High cloud cover | Dimensionless | The proportion of a grid box covered by cloud occurring in the high levels of the troposphere. High cloud is a single-level field calculated from cloud occurring on model levels with a pressure less than 0.45 times the surface pressure. So, if the surface pressure is 1000 hPa (hectopascal), high cloud would be calculated using levels with a pressure of less than 450 hPa (approximately 6km and above (assuming a "standard atmosphere")). The high cloud cover parameter is calculated from the cloud for the appropriate model levels described above. Assumptions are made about the degree of overlap/randomness between clouds in different model levels. Cloud fractions vary from 0 to 1. | 0.224129 |

| Low cloud cover | Dimensionless | This parameter is the proportion of a grid box covered by cloud occurring in the lower levels of the troposphere. Low cloud is a single level field calculated from cloud occurring on model levels with a pressure greater than 0.8 times the surface pressure. So, if the surface pressure is 1000 hPa (hectopascal), low cloud would be calculated using levels with a pressure greater than 800 hPa (below approximately 2km (assuming a "standard atmosphere")). Assumptions are made about the degree of overlap/randomness between clouds in different model levels. This parameter has values from 0 to 1. | 0.000000 |
|---|---|---|---|
| Gravitational potential energy | $m^2\ s^{-2}$ | This parameter is the gravitational potential energy of a unit mass, at a particular location at the surface of the Earth, relative to mean sea level. It is also the amount of work that would have to be done, against the force of gravity, to lift a unit mass to that location from mean sea level. The (surface) geopotential height (orography) can be calculated by dividing the (surface) geopotential by the Earth's gravitational acceleration, g (=9.80665 m s-2 ). This parameter does not vary in time. | NaN |
| Medium cloud cover | Dimensionless | This parameter is the proportion of a grid box covered by cloud occurring in the middle levels of the troposphere. Medium cloud is a single level field calculated from cloud occurring on model levels with a pressure between 0.45 and 0.8 times the surface pressure. So, if the surface pressure is 1000 hPa (hectopascal), medium cloud would be calculated using levels with a pressure of less than or equal to 800 hPa and greater than or equal to 450 hPa (between approximately 2km and 6km (assuming a "standard atmosphere")). The medium cloud parameter is calculated from cloud cover for the appropriate model levels as described above. Assumptions are made about the degree of overlap/randomness between clouds in different model levels. Cloud fractions vary from 0 to 1. | 0.000000 |

| Maximum temperature | k | This parameter is the highest temperature of air at 2m above the surface of land, sea or inland water since the parameter was last archived in a particular forecast. 2m temperature is calculated by interpolating between the lowest model level and the Earth's surface, taking account of the atmospheric conditions. This parameter has units of kelvin (K). Temperature measured in kelvin can be converted to degrees Celsius (°C) by subtracting 273.15. | 273.357666 |
|---|---|---|---|
| Maximum precipitation rate | $kg\ m^{-2}\ s^{-1}$ | The total precipitation is calculated from the combined large-scale and convective rainfall and snowfall rates every time step and the maximum is kept since the last postprocessing. | 0.000000 |
| Mean sea level pressure | Pa | This parameter is the pressure (force per unit area) of the atmosphere at the surface of the Earth, adjusted to the height of mean sea level. It is a measure of the weight that all the air in a column vertically above a point on the Earth's surface would have, if the point were located at mean sea level. It is calculated over all surfaces - land, sea and inland water. Maps of mean sea level pressure are used to identify the locations of low and high pressure weather systems, often referred to as cyclones and anticyclones. Contours of mean sea level pressure also indicate the strength of the wind. Tightly packed contours show stronger winds. The units of this parameter are pascals (Pa). Mean sea level pressure is often measured in hPa and sometimes is presented in the old units of millibars, mb (1 hPa = 1 mb = 100 Pa). | 101550.976562 |
| Minimum temperature | k | This parameter is the lowest temperature of air at 2m above the surface of land, sea or inland waters since the parameter was last archived in a particular forecast. 2m temperature is calculated by interpolating between the lowest model level and the Earth's surface, taking account of the atmospheric conditions. See further information. This parameter has units of kelvin (K). Temperature measured in kelvin can be converted to degrees Celsius (°C) by subtracting 273.15. | 273.357666 |
| Minimum precipitation rate | $kg\ m^{-2}\ s^{-1}$ | The total precipitation is calculated from the combined large-scale and convective rainfall and snowfall rates every time step and the minimum is kept since the last postprocessing. | 0.000000 |

| | | | |
|---|---|---|---|
| Precipitation type | Dimensionless | This parameter describes the type of precipitation at the surface, at the specified time. A precipitation type is assigned wherever there is a non-zero value of precipitation. The ECMWF Integrated Forecasting System (IFS) has only two predicted precipitation variables: rain and snow. Precipitation type is derived from these two predicted variables in combination with atmospheric conditions, such as temperature. Values of precipitation type defined in the IFS: 0: No precipitation, 1: Rain, 3: Freezing rain (i.e. supercooled raindrops which freeze on contact with the ground and other surfaces), 5: Snow, 6: Wet snow (i.e. snow particles which are starting to melt); 7: Mixture of rain and snow, 8: Ice pellets. These precipitation types are consistent with WMO Code Table 4.201. Other types in this WMO table are not defined in the IFS. | 0.000000 |
| Rain water content of atmosphere | $kg\ m^{-2}$ | This parameter is the total amount of water in droplets of raindrop size (which can fall to the surface as precipitation) in a column extending from the surface of the Earth to the top of the atmosphere. This parameter represents the area averaged value for a grid box. Clouds contain a continuum of different sized water droplets and ice particles. The ECMWF Integrated Forecasting System (IFS) cloud scheme simplifies this to represent a number of discrete cloud droplets/particles including: cloud water droplets, raindrops, ice crystals and snow (aggregated ice crystals). Droplet formation, conversion and aggregation processes are also highly simplified in the IFS. 0.000000 | 0.000000 |
| Snow density | $kg\ m^{-3}$ | This parameter is the mass of snow per cubic metre in the snow layer. The ECMWF Integrated Forecasting System (IFS) represents snow as a single additional layer over the uppermost soil level. The snow may cover all or part of the grid box. This parameter is defined over the whole globe, even where there is no snow. Regions without snow can be masked out by only considering grid points where the snow depth (m of water equivalent) is greater than 0.0. | 99.999985 |

| Snow depth | m of water equivalent | This parameter is the amount of snow from the snow-covered area of a grid box. Its units are metres of water equivalent, so it is the depth the water would have if the snow melted and was spread evenly over the whole grid box. The ECMWF Integrated Forecasting System (IFS) represents snow as a single additional layer over the uppermost soil level. The snow may cover all or part of the grid box. | 0.000000 |
| Snowfall | m of water equivalent | This parameter is the accumulated snow that falls to the Earth's surface. It is the sum of large-scale snowfall and convective snowfall. Large-scale snowfall is generated by the cloud scheme in the ECMWF Integrated Forecasting System (IFS). The cloud scheme represents the formation and dissipation of clouds and large-scale precipitation due to changes in atmospheric quantities (such as pressure, temperature and moisture) predicted directly at spatial scales of the grid box or larger. Convective snowfall is generated by the convection scheme in the IFS, which represents convection at spatial scales smaller than the grid box. In the IFS, precipitation is comprised of rain and snow. This parameter is accumulated over a particular time period which depends on the data extracted. For the reanalysis, the accumulation period is over the 1 hour ending at the validity date and time. For the ensemble members, ensemble mean and ensemble spread, the accumulation period is over the 3 hours ending at the validity date and time. The units of this parameter are depth in metres of water equivalent. It is the depth the water would have if it were spread evenly over the grid box. Care should be taken when comparing model parameters with observations, because observations are often local to a particular point in space and time, rather than representing averages over a model grid box. | 0.000000 |

| Soil temperature (0 to 7 cm) | K | This parameter is the temperature of the soil at level 1 (in the middle of layer 1). The ECMWF Integrated Forecasting System (IFS) has a four-layer representation of soil, where the surface is at 0cm: Layer 1: 0 - 7cm, Layer 2: 7 - 28cm, Layer 3: 28 - 100cm, Layer 4: 100 - 289cm. Soil temperature is set at the middle of each layer, and heat transfer is calculated at the interfaces between them. It is assumed that there is no heat transfer out of the bottom of the lowest layer. Soil temperature is defined over the whole globe, even over ocean. Regions with a water surface can be masked out by only considering grid points where the land-sea mask has a value greater than 0.5. This parameter has units of kelvin (K). Temperature measured in kelvin can be converted to degrees Celsius (°C) by subtracting 273.15. | 276.865784 |
|---|---|---|---|
| Soil temperature (7 to 28 cm) | K | This parameter is the temperature of the soil at level 2 (in the middle of layer 2). The ECMWF Integrated Forecasting System (IFS) has a four-layer representation of soil, where the surface is at 0cm: Layer 1: 0 - 7cm, Layer 2: 7 - 28cm, Layer 3: 28 - 100cm, Layer 4: 100 - 289cm. Soil temperature is set at the middle of each layer, and heat transfer is calculated at the interfaces between them. It is assumed that there is no heat transfer out of the bottom of the lowest layer. Soil temperature is defined over the whole globe, even over ocean. Regions with a water surface can be masked out by only considering grid points where the land-sea mask has a value greater than 0.5. This parameter has units of kelvin (K). Temperature measured in kelvin can be converted to degrees Celsius (°C) by subtracting 273.15. | 282.708038 |

| Soil temperature (28 to 100 cm) | K | This parameter is the temperature of the soil at level 3 (in the middle of layer 3). The ECMWF Integrated Forecasting System (IFS) has a four-layer representation of soil, where the surface is at 0cm: Layer 1: 0 - 7cm, Layer 2: 7 - 28cm, Layer 3: 28 - 100cm, Layer 4: 100 - 289cm. Soil temperature is set at the middle of each layer, and heat transfer is calculated at the interfaces between them. It is assumed that there is no heat transfer out of the bottom of the lowest layer. Soil temperature is defined over the whole globe, even over ocean. Regions with a water surface can be masked out by only considering grid points where the land-sea mask has a value greater than 0.5. This parameter has units of kelvin (K). Temperature measured in kelvin can be converted to degrees Celsius (°C) by subtracting 273.15. | 286.920227 |
|---|---|---|---|
| Soil temperature (100 to 289 cm) | K | This parameter is the temperature of the soil at level 4 (in the middle of layer 4). The ECMWF Integrated Forecasting System (IFS) has a four-layer representation of soil, where the surface is at 0cm: Layer 1: 0 - 7cm, Layer 2: 7 - 28cm, Layer 3: 28 - 100cm, Layer 4: 100 - 289cm. Soil temperature is set at the middle of each layer, and heat transfer is calculated at the interfaces between them. It is assumed that there is no heat transfer out of the bottom of the lowest layer. Soil temperature is defined over the whole globe, even over ocean. Regions with a water surface can be masked out by only considering grid points where the land-sea mask has a value greater than 0.5. This parameter has units of kelvin (K). Temperature measured in kelvin can be converted to degrees Celsius (°C) by subtracting 273.15. | 290.265320 |
| Snow water content of atmosphere | $k\,m^{-2}$ | This parameter is the total amount of water in the form of snow (aggregated ice crystals which can fall to the surface as precipitation) in a column extending from the surface of the Earth to the top of the atmosphere. This parameter represents the area averaged value for a grid box. Clouds contain a continuum of different sized water droplets and ice particles. The ECMWF Integrated Forecasting System (IFS) cloud scheme simplifies this to represent a number of discrete cloud droplets/particles including: cloud water droplets, raindrops, ice crystals and snow (aggregated ice crystals). Droplet formation, conversion and aggregation processes are also highly simplified in the IFS. | 0.000069 |

| Soil water (0 to 7 cm) | $m^3 m^{-3}$ | This parameter is the volume of water in soil layer 1 (0 - 7cm, the surface is at 0cm). The ECMWF Integrated Forecasting System (IFS) has a four-layer representation of soil: Layer 1: 0 - 7cm, Layer 2: 7 - 28cm, Layer 3: 28 - 100cm, Layer 4: 100 - 289cm. Soil water is defined over the whole globe, even over ocean. Regions with a water surface can be masked out by only considering grid points where the land-sea mask has a value greater than 0.5. The volumetric soil water is associated with the soil texture (or classification), soil depth, and the underlying groundwater level. | 0.439442 |
| Soil water (7 to 28 cm) | $m^3 m^{-3}$ | This parameter is the volume of water in soil layer 2 (7 - 28cm, the surface is at 0cm). The ECMWF Integrated Forecasting System (IFS) has a four-layer representation of soil: Layer 1: 0 - 7cm, Layer 2: 7 - 28cm, Layer 3: 28 - 100cm, Layer 4: 100 - 289cm. Soil water is defined over the whole globe, even over ocean. Regions with a water surface can be masked out by only considering grid points where the land-sea mask has a value greater than 0.5. The volumetric soil water is associated with the soil texture (or classification), soil depth, and the underlying groundwater level. | 0.447512 |
| Soil water (28 to 100 cm) | $m^3 m^{-3}$ | This parameter is the volume of water in soil layer 3 (28 - 100cm, the surface is at 0cm). The ECMWF Integrated Forecasting System (IFS) has a four-layer representation of soil: Layer 1: 0 - 7cm, Layer 2: 7 - 28cm, Layer 3: 28 - 100cm, Layer 4: 100 - 289cm. Soil water is defined over the whole globe, even over ocean. Regions with a water surface can be masked out by only considering grid points where the land-sea mask has a value greater than 0.5. The volumetric soil water is associated with the soil texture (or classification), soil depth, and the underlying groundwater level. | 0.387898 |

| | | | |
|---|---|---|---|
| Soil water (100 to 289 cm) | $m^3m^{-3}$ | This parameter is the volume of water in soil layer 4 (100 - 289cm, the surface is at 0cm). The ECMWF Integrated Forecasting System (IFS) has a four-layer representation of soil: Layer 1: 0 - 7cm, Layer 2: 7 - 28cm, Layer 3: 28 - 100cm, Layer 4: 100 - 289cm. Soil water is defined over the whole globe, even over ocean. Regions with a water surface can be masked out by only considering grid points where the land-sea mask has a value greater than 0.5. The volumetric soil water is associated with the soil texture (or classification), soil depth, and the underlying groundwater level. | 0.380035 |
| Solar radiation | $Jm^{-2}$ | This parameter is the amount of solar radiation (also known as shortwave radiation) that reaches a horizontal plane at the surface of the Earth. This parameter comprises both direct and diffuse solar radiation. Radiation from the Sun (solar, or shortwave, radiation) is partly reflected back to space by clouds and particles in the atmosphere (aerosols) and some of it is absorbed. The rest is incident on the Earth's surface (represented by this parameter). To a reasonably good approximation, this parameter is the model equivalent of what would be measured by a pyranometer (an instrument used for measuring solar radiation) at the surface. However, care should be taken when comparing model parameters with observations, because observations are often local to a particular point in space and time, rather than representing averages over a model grid box. This parameter is accumulated over a particular time period which depends on the data extracted. The units are joules per square metre (J m-2). To convert to watts per square metre (W m-2), the accumulated values should be divided by the accumulation period expressed in seconds. The ECMWF convention for vertical fluxes is positive downwards. | 0.000000 |
| Solar radiation (clear sky) | $Jm^{-2}$ | Clear-sky downward shortwave radiation flux at surface computed from the model radiation scheme. | 0.000000 |

| | | | |
|---|---|---|---|
| Solar radiation (top of atmosphere) | $Jm^{-2}$ | This parameter is the incoming solar radiation (also known as shortwave radiation) minus the outgoing solar radiation at the top of the atmosphere. It is the amount of radiation passing through a horizontal plane. The incoming solar radiation is the amount received from the Sun. The outgoing solar radiation is the amount reflected and scattered by the Earth's atmosphere and surface.
This parameter is accumulated over a particular time period which depends on the data extracted. The units are joules per square metre (J m-2). To convert to watts per square metre (W m-2), the accumulated values should be divided by the accumulation period expressed in seconds.
The ECMWF convention for vertical fluxes is positive downwards | 0.000000 |
| Solar radiation (total sky) | $J\,m^{-2}$ | This parameter is the amount of solar (shortwave) radiation reaching the surface of the Earth (both direct and diffuse) minus the amount reflected by the Earth's surface (which is governed by the albedo), assuming clear-sky (cloudless) conditions. It is the amount of radiation passing through a horizontal plane. Clear-sky radiation quantities are computed for exactly the same atmospheric conditions of temperature, humidity, ozone, trace gases and aerosol as the corresponding total-sky quantities (clouds included), but assuming that the clouds are not there. Radiation from the Sun (solar, or shortwave, radiation) is partly reflected back to space by clouds and particles in the atmosphere (aerosols) and some of it is absorbed. The rest is incident on the Earth's surface, where some of it is reflected. The difference between downward and reflected solar radiation is the surface net solar radiation. This parameter is accumulated over a particular time period which depends on the data extracted. For the reanalysis, the accumulation period is over the 1 hour ending at the validity date and time. For the ensemble members, ensemble mean and ensemble spread, the accumulation period is over the 3 hours ending at the validity date and time. The units are joules per square metre (J m-2 ). To convert to watts per square metre (W m-2 ), the accumulated values should be divided by the accumulation period expressed in seconds. The ECMWF convention for vertical fluxes is positive downwards. | 0.000000 |

| Solar radiation (top of atmosphere) (clear sky) | J m$^{-2}$ | This parameter is the incoming solar radiation (also known as shortwave radiation) minus the outgoing solar radiation at the top of the atmosphere, assuming clear-sky (cloudless) conditions. It is the amount of radiation passing through a horizontal plane. The incoming solar radiation is the amount received from the Sun. The outgoing solar radiation is the amount reflected and scattered by the Earth's atmosphere and surface, assuming clear-sky (cloudless) conditions. Clear-sky radiation quantities are computed for exactly the same atmospheric conditions of temperature, humidity, ozone, trace gases and aerosol as the total-sky (clouds included) quantities, but assuming that the clouds are not there. This parameter is accumulated over a particular time period which depends on the data extracted. For the reanalysis, the accumulation period is over the 1 hour ending at the validity date and time. For the ensemble members, ensemble mean and ensemble spread, the accumulation period is over the 3 hours ending at the validity date and time. The units are joules per square metre (J m-2 ). To convert to watts per square metre (W m-2 ), the accumulated values should be divided by the accumulation period expressed in seconds. The ECMWF convention for vertical fluxes is positive downwards. | 0.000000 |
| Temperature | K | This parameter is the temperature in the atmosphere. It has units of kelvin (K). Temperature measured in kelvin can be converted to degrees Celsius (°C) by subtracting 273.15. | 272.976929 |
| Surface pressure | Pa | This parameter is the pressure (force per unit area) of the atmosphere at the surface of land, sea and inland water. It is a measure of the weight of all the air in a column vertically above a point on the Earth's surface. Surface pressure is often used in combination with temperature to calculate air density. The strong variation of pressure with altitude makes it difficult to see the low and high pressure weather systems over mountainous areas, so mean sea level pressure, rather than surface pressure, is normally used for this purpose. The units of this parameter are Pascals (Pa). Surface pressure is often measured in hPa and sometimes is presented in the old units of millibars, mb (1 hPa = 1 mb= 100 Pa). | 99115.242188 |

| Thermal radiation | $Jm^{-2}$ | This parameter is the amount of thermal (also known as longwave or terrestrial) radiation emitted by the atmosphere and clouds that reaches a horizontal plane at the surface of the Earth. The surface of the Earth emits thermal radiation, some of which is absorbed by the atmosphere and clouds. The atmosphere and clouds likewise emit thermal radiation in all directions, some of which reaches the surface (represented by this parameter). This parameter is accumulated over a particular time period which depends on the data extracted. The units are joules per square metre (J m-2). To convert to watts per square metre (W m-2), the accumulated values should be divided by the accumulation period expressed in seconds. | 845375.562500 |
| Thermal radiation (clear sky) | $Jm^{-2}$ | Clear-sky downward longwave radiation flux at surface computed from the model radiation scheme. | 849147.312500 |
| Thermal radiation (top of atmosphere) | $J\,m^{-2}$ | The thermal (also known as terrestrial or longwave) radiation emitted to space at the top of the atmosphere is commonly known as the Outgoing Longwave Radiation (OLR). The top net thermal radiation (this parameter) is equal to the negative of OLR. This parameter is accumulated over a particular time period which depends on the data extracted. For the reanalysis, the accumulation period is over the 1 hour ending at the validity date and time. For the ensemble members, ensemble mean and ensemble spread, the accumulation period is over the 3 hours ending at the validity date and time. The units are joules per square metre (J m-2 ). To convert to watts per square metre (W m-2 ), the accumulated values should be divided by the accumulation period expressed in seconds. The ECMWF convention for vertical fluxes is positive downwards. | -854573.250000 |

| | | | |
|---|---|---|---|
| Thermal radiation (top of atmosphere) (clear sky) | J m$^{-2}$ | This parameter is the thermal (also known as terrestrial or longwave) radiation emitted to space at the top of the atmosphere, assuming clear-sky (cloudless) conditions. It is the amount passing through a horizontal plane. Note that the ECMWF convention for vertical fluxes is positive downwards, so a flux from the atmosphere to space will be negative. Clear-sky radiation quantities are computed for exactly the same atmospheric conditions of temperature, humidity, ozone, trace gases and aerosol as total-sky quantities (clouds included), but assuming that the clouds are not there. The thermal radiation emitted to space at the top of the atmosphere is commonly known as the Outgoing Longwave Radiation (OLR) (i.e., taking a flux from the atmosphere to space as positive). Note that OLR is typically shown in units of watts per square metre (W m-2 ). This parameter is accumulated over a particular time period which depends on the data extracted. For the reanalysis, the accumulation period is over the 1 hour ending at the validity date and time. For the ensemble members, ensemble mean and ensemble spread, the accumulation period is over the 3 hours ending at the validity date and time. The units are joules per square metre (J m-2 ). To convert to watts per square metre (W m-2 ), the accumulated values should be divided by the accumulation period expressed in seconds. | -853921.937500 |
| Total cloud cover | Dimensionless | This parameter is the proportion of a grid box covered by cloud. Total cloud cover is a single level field calculated from the cloud occurring at different model levels through the atmosphere. Assumptions are made about the degree of overlap/randomness between clouds at different heights. Cloud fractions vary from 0 to 1. | 0.224129 |

| Total precipitation | m | This parameter is the accumulated liquid and frozen water, comprising rain and snow, that falls to the Earth's surface. It is the sum of large-scale precipitation and convective precipitation. Large-scale precipitation is generated by the cloud scheme in the ECMWF Integrated Forecasting System (IFS). The cloud scheme represents the formation and dissipation of clouds and large-scale precipitation due to changes in atmospheric quantities (such as pressure, temperature and moisture) predicted directly by the IFS at spatial scales of the grid box or larger. Convective precipitation is generated by the convection scheme in the IFS, which represents convection at spatial scales smaller than the grid box. This parameter does not include fog, dew or the precipitation that evaporates in the atmosphere before it lands at the surface of the Earth. This parameter is accumulated over a particular time period which depends on the data extracted. For the reanalysis, the accumulation period is over 1 hour, ending at the validity date and time. For the ensemble members, ensemble mean and ensemble spread, the accumulation period is over the 3 hours ending at the validity date and time. The units of this parameter are depth in metres of water equivalent. It is the depth the water would have if it were spread evenly over the grid box. Care should be taken when comparing model parameters with observations, because observations are often local to a particular point in space and time, rather than representing averages over a model grid box. | 0.000000 |

