# OpenReview forum: "Tensor Time-Series Forecasting and Anomaly Detection with Augmented Causality"
_ICLR.cc/2024/Conference — Submitted to ICLR 2024_

### Official Review · Reviewer_TMqL · 2023-10-28

**Soundness:** 2 fair
**Presentation:** 2 fair
**Contribution:** 1 poor
**Rating:** 3
**Confidence:** 4

**Summary:**

In this manuscript, the authors start from the tensor time series, which can encode higher dimensional information than classic multivariate time series, and aim to discover and leverage their fine-grained time-dependent causal relations to contribute to a more accurate analysis. To this end, the authors first form an augmented Granger Causality model, named TBN-Granger Causality, which adds time-respecting Bayesian Networks to the time-lagged Neural Granger Causality through a bi-level optimization, such that the overlooking of instantaneous effects in typical causal time series analysis can be addressed.

**Strengths:**

1. Compared with only three baseline models, the proposed solution is significantly better. In particular, a unique visual example is given to analyze the effectiveness of the proposed method.

**Weaknesses:**

1. The authors ignore a large number of recent excellent graph convolution or graph attention models for time series prediction. It is worth mentioning that the classical baseline DCRNN is the result of 2018 years of reporting and is almost outperformed by current solutions. Obviously, it is difficult for the authors to complete the comparison of SOTA algorithm in such a short rebuttal period. Therefore, I believe that the experimental results of the manuscript cannot fully illustrate the effectiveness of the proposed method.

Interestingly, recent studies [1-2] in spatiotemporal representation learning (from the perspective of the equivariance [1] and latent fields [2]) have investigated the common characteristics among for related tasks, such as traffic, physical simulations, motion tracking.

[1] Miltiadis Kofinas, et al., 2021. Roto-translated Local Coordinate Frames For Interacting Dynamical Systems. In Proceedings of the NeurIPS, pp. 6417-6429.
[2] Miltiadis Kofinas, et al., 2023. Latent Field Discovery in Interacting Dynamical Systems with Neural Fields. In Proceedings of the NeurIPS.

From the perspective of graph convolutional networks for Spatio-Temporal representation learning, some baselines of current excellent performance are listed as [3-6]. Although these studies focus on traffic flow or smart city studies, they are similar to meteorological information in different geographical locations from the perspective of geospatial information mining.

E.g., STGCN [3], STSGCN [4], STFGNN [5] and ST-SSL [6].
[3] Yu, B.; Yin, H.; and Zhu, Z. 2018. Spatio-Temporal Graph Convolutional Networks: A Deep Learning Framework for Traffc Forecasting. In Proceedings of the Twenty-Seventh International Joint Conference on Artifcial Intelligence, IJCAI, pp. 3634–3640.
[4] Song, C.; Lin, Y.; Guo, S.; and Wan, H. 2020. Spatial-Temporal Synchronous Graph Convolutional Networks: A New Framework for Spatial-Temporal Network Data Forecasting. In the Thirty-Fourth AAAI Conference on Artificial Intelligence, AAAI, pp. 914-921.
[5] Li, M.; and Zhu, Z. 2021. Spatial-temporal fusion graph neural networks for traffc fow forecasting. In Proceedings of the AAAI Conference on Artifcial Intelligence, pp. 4189–4196.
[6] Jiahao Ji, et al., 2023. Spatio-Temporal Self-Supervised Learning for Traffic Flow Prediction. In Proceedings of the AAAI Conference on Artifcial Intelligence, pp. 4356-4364.

The above comments are expected to be revised and discussed by the authors. If the rebuttal period cannot be revised, it also hopes to help you improve the quality of the manuscript.

**Questions:**

Please see details of the weaknesses.

---

> ### Author Response · Authors · 2023-11-23
> **Addressing the concerns of Reviewer TMqL**
>
> Thanks for the review and paying attention to our forecasting module. Your concerns are listed and addressed below.
>
> * First of all, **all the suggested literature [1-6] is added and discussed in the revised paper and colored blue**. Furthermore, paper [2] is a very new paper accepted by NeurIPS 2023 and has not been released to the public yet. Then, we discussed its content based on the previous version on arXiv.
>
> - Second, we add state-of-the-art baselines since one of our baselines, DCRNN, is 2018. On the one hand, we would like to mention our other baseline GTS is in ICLR 2021, which is newer than the suggested baselines [3] in IJCAI 2018, [4] in AAAI 2020, and [5] in AAAI 2021. On the other hand, in the suggested SOTA algorithm [6] in AAAI 2023, baselines [3-5] are included and compared. **Therefore, we did the new experiments for the SOTA baseline ST-SSL [6].** The new table is shown below and has also been included in the revised paper (colored blue). In general, the SOTA baseline ST-SSL achieves a very competitive forecasting performance through contrastive learning, which is superior to one of our proposed methods TacSas, and slightly lower than our TacSas++, which is newly proposed by not only discovering the fine-grained temporal structures but also getting calibrated by contrasting the nearest time window for persistence forecasting.
>
> |            | ERA5-2017 $(\downarrow)$ | ERA5-2018 $(\downarrow)$ | ERA5-2019 $(\downarrow)$ | ERA5-2020 $(\downarrow)$ |
> |------------|--------------------------|--------------------------|--------------------------|--------------------------|
> | GRU        | 1.8834 $\pm$ 0.0126       | 1.9764 $\pm$ 0.1466       | 1.6194 $\pm$ 0.2645       | 1.7859 $\pm$ 0.2324       |
> | DCRNN      | 0.0819 $\pm$ 0.0025       | 0.0797 $\pm$ 0.0049       | 0.0799 $\pm$ 0.0035       | 0.0826 $\pm$ 0.0033       |
> | GTS        | 0.0777 $\pm$ 0.0054       | 0.0766 $\pm$ 0.0029       | 0.0760 $\pm$ 0.0031       | 0.0742 $\pm$ 0.0021       |
> | TacSas     | 0.0496 $\pm$ 0.0017       | 0.0499 $\pm$ 0.0017       | 0.0502 $\pm$ 0.0016       | 0.0488 $\pm$ 0.0019       |
> | ST-SSL     | 0.0345 $\pm$ 0.0051      | 0.0330 $\pm$ 0.0018       | 0.0361 $\pm$ 0.0021       | 0.0348 $\pm$ 0.0020       |
> | TacSas++    | $\textbf{0.0271}$ $\pm$ $\textbf{0.0004}$ | $\textbf{0.0276}$ $\pm$ $\textbf{0.0004}$ | $\textbf{0.0282}$ $\pm$ $\textbf{0.0003}$ | $\textbf{0.0265}$ $\pm$ $\textbf{0.0004}$ |
>
>
> ST-SSL relies on the off-the-shelf input structure like what DCRNN does, but its input is usually time-evolving and hardly acquired in most real-world settings (and usually gets replaced by random ones). It is this very problem that motivates our paper.
>
> * Third, thanks again for your raised concern on our forecasting module. **They are very actionable and improve our paper**. **Please also do not overlook our theoretical and empirical efforts** in terms of **fine-grained temporal-spatial structures discovery**, **its leverage for anomaly detection**, **the end-to-end generative model for realizing them all**, and **the new data source**.

---

### Official Review · Reviewer_kqnD · 2023-10-31

**Soundness:** 3 good
**Presentation:** 3 good
**Contribution:** 2 fair
**Rating:** 3
**Confidence:** 3

**Summary:**

The paper proposed a model for spatial-temporal learning for forecasting and anomaly detection in weather prediction. The proposed model finds embedding for location and uses causal modeling to model the relationship between location for the final task. As experimental evidence paper shows results on 2 data sets and a comparison with only 3 benchmark models.

**Strengths:**

The paper is technically sound. I am aware of many spatio-temporal models but not sure if the causal model has been explored previously for this task. To me it paper gives a novel probabilistic model for the spatio temporal learning.

The paper is well-written and easy to follow.

**Weaknesses:**

Literature Survey misses many SOTA for example

Spatio-temporal forecasting :
1. Li, Peiyuan, Yin Yu, Daning Huang, Zhi‐Hua Wang, and Ashish Sharma. "Regional heatwave prediction using Graph Neural Network and weather station data." Geophysical Research Letters 50, no. 7 (2023): e2023GL103405.
Anomaly detection
2. Lira, Hernan, Luis Martí, and Nayat Sanchez-Pi. "A graph neural network with spatio-temporal attention for multi-sources time series data: An application to frost forecast." Sensors 22, no. 4 (2022): 1486.

The author should include more recent literature in his survey.

According to my understanding, this approach only detects if some anomaly has a very high value, i.e.,  it detected only extreme values. But. In time series temporal anomalies i.e., anomalies containing the rare temporal pattern, and inter-variable i.e. anomalies where the inter-variable relationship does not match with the common pattern are of main concern. The paper did not detect. It needs to show the performance of the proposed model in various data in terms of temporal, inter-variable, or inter-location anomalies too. Please check

Zhang, C., Song, D., Chen, Y., Feng, X., Lumezanu, C., Cheng, W., Ni, J., Zong, B., Chen, H. and Chawla, N.V., 2019, July. A deep neural network for unsupervised anomaly detection and diagnosis in multivariate time series data. In Proceedings of the AAAI conference on artificial intelligence (Vol. 33, No. 01, pp. 1409-1416).

**Questions:**

Why author did not compare with sota in the case of forecasting?
1. Li, Peiyuan, Yin Yu, Daning Huang, Zhi‐Hua Wang, and Ashish Sharma. "Regional heatwave prediction using Graph Neural Network and weather station data." Geophysical Research Letters 50, no. 7 (2023): e2023GL103405.
Anomaly detection
2. Lira, Hernan, Luis Martí, and Nayat Sanchez-Pi. "A graph neural network with spatio-temporal attention for multi-sources time series data: An application to frost forecast." Sensors 22, no. 4 (2022): 1486.

Why author did not go for other anomalies other than extreme value?

---

> ### Author Response · Authors · 2023-11-23
> **Addressing the concerns of Reviewer kqnD**
>
> Thanks so much for your questions and concerns, they are listed and addressed below.
>
> * First, **all your suggested related work is added and discussed in the updated paper and colored blue**.
>
> - Second, the reason why we did not compare with your suggested SOTA baseline in the case of forecasting "Li et al., Geophysical Research Letters 50" and "Lira et al, Sensors 22" is that we searched online but **did find the published code** for those methods. But, on the one hand, they are now added to the revised paper and discussed. On the other hand, **we added a new sota baseline** suggested by Reviewer TMqL for forecasting, which is a new paper this year "Ji et al., AAAI 2023. Spatio-Temporal Self-Supervised Learning for Traffic Flow Prediction.". In short, our proposed method can still outperform the baselines as follows. More details are added in the revised paper.
>
> |            | ERA5-2017 $(\downarrow)$ | ERA5-2018 $(\downarrow)$ | ERA5-2019 $(\downarrow)$ | ERA5-2020 $(\downarrow)$ |
> |------------|--------------------------|--------------------------|--------------------------|--------------------------|
> | GRU        | 1.8834 $\pm$ 0.0126       | 1.9764 $\pm$ 0.1466       | 1.6194 $\pm$ 0.2645       | 1.7859 $\pm$ 0.2324       |
> | DCRNN      | 0.0819 $\pm$ 0.0025       | 0.0797 $\pm$ 0.0049       | 0.0799 $\pm$ 0.0035       | 0.0826 $\pm$ 0.0033       |
> | GTS        | 0.0777 $\pm$ 0.0054       | 0.0766 $\pm$ 0.0029       | 0.0760 $\pm$ 0.0031       | 0.0742 $\pm$ 0.0021       |
> | TacSas     | 0.0496 $\pm$ 0.0017       | 0.0499 $\pm$ 0.0017       | 0.0502 $\pm$ 0.0016       | 0.0488 $\pm$ 0.0019       |
> | ST-SSL     | 0.0345 $\pm$ 0.0051      | 0.0330 $\pm$ 0.0018       | 0.0361 $\pm$ 0.0021       | 0.0348 $\pm$ 0.0020       |
> | TacSas++    | $\textbf{0.0271}$ $\pm$ $\textbf{0.0004}$ | $\textbf{0.0276}$ $\pm$ $\textbf{0.0004}$ | $\textbf{0.0282}$ $\pm$ $\textbf{0.0003}$ | $\textbf{0.0265}$ $\pm$ $\textbf{0.0004}$ |
>
> * Third, "_why did not consider other kinds of anomalies, like inter-variable relationship anomalies_"? Thanks for the question. To the best of our knowledge, facing various types of anomalies, different methods are proposed for different anomaly patterns. On the one hand, in our paper, the proposed model did the value anomaly detection, and the selected baselines and discussed related work are also the methods that only did the value anomaly detection. On the other hand, the structural anomaly detection method you suggested also targets one specific type of anomaly among various anomaly patterns, which is also unable to do value anomaly detection. Each aspect has its own value, like value anomaly detection can help with extreme weather detection, and structural anomaly detection can help with high-tech equipment deployment. **Proposing a universal anomaly detection method for all possible anomaly patterns is very interesting, we would like to explore it as a future direction. But for now, it is beyond the scope of our submission.**
>
> * Finally, thanks again for your raised concerns. **Please also do not overlook our theoretical and empirical efforts** in terms of **fine-grained temporal-spatial structures discovery**, **its leverage for anomaly detection**, **the end-to-end generative model for realizing them all**, and **the new data source**.

---

### Official Review · Reviewer_TM7t · 2023-10-31

**Soundness:** 1 poor
**Presentation:** 2 fair
**Contribution:** 1 poor
**Rating:** 1
**Confidence:** 3

**Summary:**

This paper introduces TacSas, an end-to-end methodology for "tensor" time series forecasting that can be used for both forecasting in earth sciences, Granger causal discovery, anomaly detection, among others. TacSas reportedly includes 1/a custom pretrained autoencoder to preprocess tensor time series, 2/ a module for causal discovery for instantaneous effects and Granger-causality across time via bi-level optimization, 3/ anomaly detection in high dimensions inspired by extreme value theory. A few baselines are compared to, where TacSas appears to outperform baselines.

**Strengths:**

The paper posts a good review of literature for models in tensor-shaped time series and recent threads on causal discovery in time series with expressive models.

**Weaknesses:**

- The paper is poorly written. It contains many spelling, grammar and other use-of-language mistakes that make it very difficult to discern the authors' intent and extremely difficult to follow the paper.
- Many important issues are handled with an off-hand approach. For example, see the citation of EVT for 'inspiration' or the expression of Thm 3.1. The theorem posits that "under standard causal discovery assumptions" the full causal graph can be recovered. Depending on what these assumptions are, this theorem may be a groundbreaking discovery. However the "proof sketch" of Thm 3.1 simply cites the source paper and does not construct any formal argument.
- Another example is the continued reference to "tensors" as being able to represent higher dimensional time series. This is the opposite of why tensors are conceptually used. Take the toy example of a 3x3 matrix and a 9-dimensional vector. Formally, there is nothing more 'high-dimensional' about the matrix.
- My final critique is on the experiment setup. The experiments contain no simple / naive baselines for forecasting or for anomaly detection. For such a complex architecture proposed, there is only a very limited ablation study to show which parts of TacSas are critical to its success.

**Questions:**

N/A

---

> ### Author Response · Authors · 2023-11-23
> **Addressing the concerns of Reviewer TM7t**
>
> Thanks for your raised concern.
>
> * First, we further formalized and polished the language of the paper, and uploaded the revised paper.
>
> - Second, the reason why we use the term "proof sketch" is that the original paper gives the proof, that's why we included the citation in our Thm 3.1.
>
> * Third, we could not agree that the "tensor time series" is meaningless. The reason why a toy example is called a toy is because it is simply created to illustrate the key idea. Take our data sets as an example, 238 (spatial locations) * 365 (days) * 45 (weather features), each dimension has its unique meaning. Of course, you can flatten it into a vector, then the information is lost. If doing so, let alone tensor time series, multivariate time series can be represented as vectors.
>
> - Finally, we add the SOTA time series forecasting baseline ST-SSL (Ji et al., 2023) in the below table, which outperformed many simple forecasting baselines like ARIMA (Kumar and Vanaiakshi 2015) and SVR (Castro-Neto et al. 2009) in their paper.
>
> |            | ERA5-2017 $(\downarrow)$ | ERA5-2018 $(\downarrow)$ | ERA5-2019 $(\downarrow)$ | ERA5-2020 $(\downarrow)$ |
> |------------|--------------------------|--------------------------|--------------------------|--------------------------|
> | GRU        | 1.8834 $\pm$ 0.0126       | 1.9764 $\pm$ 0.1466       | 1.6194 $\pm$ 0.2645       | 1.7859 $\pm$ 0.2324       |
> | DCRNN      | 0.0819 $\pm$ 0.0025       | 0.0797 $\pm$ 0.0049       | 0.0799 $\pm$ 0.0035       | 0.0826 $\pm$ 0.0033       |
> | GTS        | 0.0777 $\pm$ 0.0054       | 0.0766 $\pm$ 0.0029       | 0.0760 $\pm$ 0.0031       | 0.0742 $\pm$ 0.0021       |
> | TacSas     | 0.0496 $\pm$ 0.0017       | 0.0499 $\pm$ 0.0017       | 0.0502 $\pm$ 0.0016       | 0.0488 $\pm$ 0.0019       |
> | ST-SSL     | 0.0345 $\pm$ 0.0051      | 0.0330 $\pm$ 0.0018       | 0.0361 $\pm$ 0.0021       | 0.0348 $\pm$ 0.0020       |
> | TacSas++    | $\textbf{0.0271}$ $\pm$ $\textbf{0.0004}$ | $\textbf{0.0276}$ $\pm$ $\textbf{0.0004}$ | $\textbf{0.0282}$ $\pm$ $\textbf{0.0003}$ | $\textbf{0.0265}$ $\pm$ $\textbf{0.0004}$ |

---

### Official Review · Reviewer_kHDs · 2023-11-06

**Soundness:** 3 good
**Presentation:** 3 good
**Contribution:** 3 good
**Rating:** 8
**Confidence:** 3

**Summary:**

This paper leverages Augmented Causality in tensor time series in forecasting and anomaly detection tasks.

It forms time-respecting Bayesian Networks to the time-lagged Neural Granger Causality (TBN-Granger Causality) model and addresses the overlooking instantaneous effects in typical causal time series analysis via a bi-level optimization.

An end-to-end deep generative model, called Time-Augmented Causal Time Series AnalysiS Model, i.e., TacSas. is proposed with experimental results showing its outperformance.

**Strengths:**

- The paper is well-written, it provides a unified yet concrete view of how to capture Granger Causality in a generative manner.
- Experimental results show consistent outperformance on both forecasting and anomaly detection.

**Weaknesses:**

- Capturing of Granger Causality in the context of tensor time series is not well reviewed in literature and methodology

**Questions:**

- The proposed method is overall interesting - However, could you provide a dedicated quantitative summary and micro case study in the Appendix regarding how TacSas can capture the ground-truth causality but also can be applied when the ground-truth causal structures are hardly available?

---

> ### Author Response · Authors · 2023-11-23
> **Addressing the concerns of Reviewer kHDs**
>
> Thanks so much for your time and review, your questions are addressed as follows.
>
> * First, **regarding how TacSas can capture the ground-truth causality**, we prepared the ground-truth benchmark Lorenz96 and compared TacSas with the SOTA baseline GVAR (Marcinke-vics and Vogt, 2021). More details can be found in Appendix B.1. In short, our TacSas can discover the ground-truth causal structure defined by Lonrenz96 with competitive accuracy compared with GVAR.
>
> - Second, **for the scenario that causal structures are hardly available**, in the climate datasets we contributed as a new time series source (where the time series items are rich but ground causal structures are not available), the visualization plotted in Figure 3 quantitatively shows the causal structures captured by TacSas, which is the potential reason why TacSas can outperform DCRNN (Li et al., 2018) and GTS (Shang et al., 2021) for capturing fine-grained temporal-spatial structures in the hardly-available scenario.
>
> * In addition, **our paper gets further improved during this rebuttal period**. We compared with another state-of-the-art baseline ST-SSL (Ji et al., 2023), proposed a more accurate forecasting module TacSas++, and extended the literature discussion for various related works. All the updates are included in the revised paper and colored blue.

---

### Meta-Review · Area_Chair_iSZ6 · 2023-12-11

**Metareview:**

The paper proposes to unifiy time-series forecasting and anomaly detection with causality, proposing an end-to-end algorithm coined TacSas. Most reviewers are critical concerning the readiness of the current submission, commenting specifically that it could be clearer and with more thorough empirical justification. The rebuttal focuses on the fact that structural anomaly detection and anomaly detection overall are two different focal points, and it should not be expected to have an algorithm that addresses them all.

I agree with the authors that the focus of the paper is to be respected. That said, given the focus on high-dimensional time-series and not simply IID populations, asking for an algorithm to detect inter-variable and temporal/over-time anomalies is not really an unreasonable request. Further, strong empirical validation is either way required for a top-tier conference like ICLR. Already by addressing comments of reviewer TMqL, a new variant of the algorithm was proposed that does improve upon baselines. However, since this new variant is sufficiently different, a resubmission is warranted.

**Justification For Why Not Higher Score:**

Simply too many things that need to be updated to be accepted.
That said, this is a promising paper and I encourage the authors to work on the clarity and validation.

**Justification For Why Not Lower Score:**

See above.

---

### Decision · Program_Chairs · 2024-01-16

Reject